# Rethinking Diffusion Model in High Dimension

## Abstract

**Curse of Dimensionality** is an unavoidable challenge in statistical probability models, yet diffusion models seem to overcome this limitation, achieving impressive results in high-dimensional data generation. Diffusion models assume that they can learn the statistical quantities of the underlying probability distribution, enabling sampling from this distribution to generate realistic samples. But is this really how they work? We argue not, based on the following observations 1) In high-dimensional sparse scenarios, the fitting target of the diffusion model's objective function degrades from a **weighted sum of multiple samples** to a **single sample**, which we believe hinders the model's ability to effectively learn essential statistical quantities such as posterior, score, or velocity field. 2) Most inference methods can **be unified within a simple framework** which involves no statistical concepts, aligns with the degraded objective function, and provides an novel and intuitive perspective on the inference process. *Code is available at Supplementary Material.*

## 1 Introduction

Diffusion models exhibit remarkable competitiveness in high-dimensional data generation scenarios, particularly in image generation Rombach et al. (2022). Beyond delivering outstanding performance, diffusion models also possess various elegant mathematical formulations. Sohl-Dickstein et al. (2015) first introduced the diffusion model approach, which utilizes a Markov chain to transform complex data distributions into simple normal distributions and then learns the posterior probability distribution corresponding to the transformation process. Ho et al. (2020) refined the objective function of diffusion models and introduced Denoised DPM. Song et al. (2020b) generalized the noise addition process of diffusion models from discrete to continuous and formulated it as a stochastic differential equation (SDE). Lipman et al. (2022) proposed a new optimization perspective based on flow matching, enabling the model to directly learn the velocity field of probability flows.

All three of the aforementioned models assume that the diffusion model can learn the statistical quantities of the data distribution. In the Markov Chain formulation, it is assumed that the model can learn the posterior probability distribution. In the SDE formulation, it is assumed that the model can learn the score of the marginal distribution. In the flow matching approach, it is assumed that the model can learn the velocity field. However, this assumption contradicts conventional understandings. Traditionally, it is believed that in high-dimensional sparse scenarios, machine learning models cannot effectively learn complex hidden probability distributions and their essential statistical quantities. This discrepancy prompts a fundamental inquiry: **This discrepancy raises a fundamental question: Do diffusion models truly learn these complex distributions and their statistical quantities as theoretically assumed? If not, why are they still able to generate high-quality samples? Could it be that diffusion models operate via a different underlying mechanism?**

**We argue that diffusion models do not learn these statistical quantities; instead, they operate via a different mechanism**.

To support this conclusion, this paper provides a detailed analysis of the objective function and inference methods of diffusion models.

Section 3 focuses on the objective function. We identify a phenomenon that emerges in high-dimensional spaces: due to data sparsity, the fitting target of the diffusion model's objective function **degrades from a weighted sum to a single sample**. Under such conditions, we argue that the

model cannot effectively learn the essential statistical quantities of the underlying data distribution, including the posterior, score, and velocity field.

Section 4 focuses on the inference method. We propose a novel inference framework that not only aligns with the degraded objective function but also unifies most existing inference methods, including DDPM Ancestral Sampling, DDIM (Song et al., 2020a), Euler, DPM-Solver (Lu et al., 2022a), DPM-Solver++ (Lu et al., 2022b), and DEIS (Zhang & Chen, 2022). Furthermore, this framework provides an entirely new and intuitive perspective for understanding the inference process, without relying on any statistical concepts.

This work makes the following key contributions:

- We present the **first rigorous analysis** of the diffusion model objective in high-dimensional sparse scenarios, demonstrating that its fitting target **degrades from a weighted sum of multiple samples to a single sample**. This degradation prevents the model from effectively capturing the underlying data distribution and its associated statistical quantities (posterior, score, velocity field).

- We further introduce a novel inference framework that **unifies most existing inference methods**, encompassing both stochastic and deterministic approaches. This framework provides an entirely new way of understanding the inference process—free from any reliance on statistical concepts—while remaining fully consistent with the degraded objective function.

- Taken together, these contributions offer a **complete and fundamentally new perspective** on high-dimensional diffusion models, covering both their training objectives and inference mechanisms. This perspective is simple, intuitive, and free from statistical concepts, opening up a promising new direction for advancing diffusion models in high-dimensional settings.

## 2 BACKGROUND

Given a batch of sampled data $X_0^0, X_0^1, \ldots, X_0^N$ from the random variable $X_0$, the diffusion model mixes the data with random noise in different proportions, forming a sequence of new variables $X_1, X_2, \cdots, X_T$. The signal-to-noise ratio (SNR), which represents the ratio of data to noise, gradually decreases, and by the final variable $X_T$, it almost consists entirely of random noise.

For the original diffusion model and VP SDE, they mix in the following way:

$$X_t = \sqrt{\bar{\alpha}_t} \cdot X_0 + \sqrt{1 - \bar{\alpha}_t} \cdot \varepsilon \tag{1}$$

where $\bar{\alpha}_t$ gradually decreases from 1 to 0, and $t$ takes discrete values from 1 to $T$.

For flow matching, it mixes in the following way:

$$X_t = (1 - \sigma_t) \cdot X_0 + \sigma_t \cdot \varepsilon \tag{2}$$

where $\sigma_t$ also gradually increases from 0 to 1, and in practice, $\sigma_t = t$ is often set. $t$ takes continuous values, $t \in [0, 1]$.

**Markov Chain-based diffusion model**   For the Markov Chain-based diffusion model, its core lies in learning the conditional posterior probability $p(x_{t-1}|x_t)$. Since the posterior probability is approximately a Gaussian function, and its variance is relatively fixed, we can focus on learning the mean of the conditional posterior probability $\mathbb{E}_{p(x_{t-1}|x_t)}[x_{t-1}]$. According to the Total Law of Expectation (Ross, 2010), the mean can be expressed in another form:

$$\mathbb{E}_{p(x_{t-1}|x_t)}[x_{t-1}] = \int p(x_0|x_t) \, E_{p(x_{t-1}|x_0,x_t)} \, (x_{t-1}) \, dx_0 \tag{3}$$

As seen from equation (7) in Ho et al. (2020), the mean of $p(x_{t-1}|x_0, x_t)$ can be expressed as a linear combination of $x_0$ and $x_t$, i.e.

$$\mathbb{E}_{p(x_{t-1}|x_0,x_t)}[x_{t-1}] = \underbrace{\frac{\sqrt{\bar{\alpha}_{t-1}}\beta_t}{1 - \bar{\alpha}_t}}_{const=C_0} \cdot x_0 + \underbrace{\frac{\sqrt{\alpha_t}(1 - \bar{\alpha}_{t-1})}{1 - \bar{\alpha}_t}}_{const=C_t} \cdot x_t \tag{4}$$

Thus, the mean of $p(x_{t-1}|x_t)$ can be further expressed as

$$\mathbb{E}_{p(x_{t-1}|x_t)}[x_{t-1}] = \int p(x_0|x_t)\left(C_0 \cdot x_0 + C_t \cdot x_t\right)dx_0 = C_0 \int p(x_0|x_t)\,x_0\,dx_0 + C_t \cdot x_t \quad (5)$$

Therefore, the objective of Markov Chain-based diffusion model can be considered as learning the mean of $p(x_0|x_t)$, i.e.

$$\min_\theta \int p(x_t)\left\|f_\theta(x_t) - \int p(x_0|x_t)\,x_0\,dx_0\right\|^2 dx_t \quad (6)$$

where $f_\theta(x_t)$ is a learnable neural network function, with input $x_t$.

**Score-based diffusion model**  For the score-based diffusion model, its core lies in learning the score of the marginal distribution $p(x_t)$ $\left(\frac{\partial \log p(x_t)}{\partial x_t}\right)$. Similar to the Markov chain-based diffusion model, by introducing another variable $X_0$, the score can be expressed in another form:

$$\frac{\partial \log p(x_t)}{\partial x_t} = \int p(x_0|x_t)\frac{\partial \log p(x_t|x_0)}{\partial x_t}dx_0 \quad (7)$$

The proof of this relationship can be found in Appendix A.2.

Since $p(x_t|x_0) \sim \mathcal{N}\left(x_t; \sqrt{\bar{\alpha}_t}x_0, \sqrt{1-\bar{\alpha}_t}\right)$, the score of $p(x_t|x_0)$ can be expressed as

$$\frac{\partial \log p(x_t|x_0)}{\partial x_t} = -\frac{x_t - \sqrt{\bar{\alpha}_t}x_0}{1-\bar{\alpha}_t} = \underbrace{\frac{\sqrt{\bar{\alpha}_t}}{1-\bar{\alpha}_t}}_{const=S_0}\cdot x_0 + \underbrace{\frac{-1}{1-\bar{\alpha}_t}}_{const=S_t}\cdot x_t \quad (8)$$

Thus, the score of $p(x_t)$ can be expressed as

$$\frac{\partial \log p(x_t)}{\partial x_t} = \int p(x_0|x_t)(S_0 \cdot x_0 + S_t \cdot x_t)\,dx_0 = S_0 \int p(x_0|x_t)\,x_0\,dx_0 + S_t \cdot x_t \quad (9)$$

Therefore, the objective of score-based diffusion model can also be considered as learning the mean of $p(x_0|x_t)$.

**Flow Matching-based diffusion model**  The core of the flow matching-based diffusion model lies in learning the velocity field of the probability flow. According to Theorem 1 in Lipman et al. (2022), the velocity field $u(x_t)$ can be expressed as a weighted sum of the conditional velocity field $u(x_t|x_0)$, i.e.

$$u(x_t) = \int p(x_0|x_t)u(x_t|x_0)dx_0 \quad (10)$$

From equation 2, we know that the conditional velocity field $u(x_t|x_0)$ is

$$u(x_t|x_0) \triangleq \frac{\mathrm{d}x_t}{\mathrm{d}t} = \varepsilon - x_0 \quad (11)$$

Thus, the velocity field $u(x_t)$ can be expressed as

$$u(x_t) = \int p(x_0|x_t)(\varepsilon - x_0)dx_0 = \varepsilon - \int p(x_0|x_t)x_0dx_0 \quad (12)$$

Therefore, the objective of flow matching-based diffusion model can also be considered as learning the mean of $p(x_0|x_t)$.

**Equivalent to predicting $X_0$**  Fitting the mean of $p(x_0|x_t)$ is equivalent to **predicting $X_0$**, i.e.

$$\min_\theta \int p(x_t)\left\|f_\theta(x_t) - \int p(x_0|x_t)\,x_0\,dx_0\right\|^2 dx_t \iff \min_\theta \iint p(x_0,x_t)\|f_\theta(x_t) - x_0\|^2\,dx_0dx_t$$

The specific proof can be found in Appendix A.1. The integrals above cannot be computed exactly and are typically approximated using Monte Carlo integration. In practice, the required samples are typically obtained via **Ancestral Sampling**. The detailed procedure is as follows: sample $X_0$ from $p(x_0)$, and then sample $X_t$ from $p(x_t|x_0)$. The pair $(X_0, X_t)$ follows the joint distribution $p(x_0, x_t)$, and the individual $X_t$ follows $p(x_t)$, and the individual $X_0$ follows $p(x_0|x_t = X_t)$.

## 3 IMPACT OF SPARSITY ON THE OBJECTIVE FUNCTION

We first show the form of the posterior probability distribution $p(x_0|x_t)$.

### 3.1 FORM OF THE POSTERIOR $p(x_0|x_t)$

For convenience, we use a unified form to represent the two mixing ways in equation 1 and equation 2 as follows: $x_t = c_0 \cdot x_0 + c_1 \cdot \varepsilon$. When $c_0^2 + c_1^2 = 1$, this represents the mixing way of Markov Chain-based and Score-based diffusion model. When $c_0 + c_1 = 1$, it represents the Flow Matching mixing way. Under this representation, $p(x_t|x_0) \sim \mathcal{N}(x_t; c_0 x_0, c_1^2)$.

From the analysis in Appendix A.3, the posterior $p(x_0|x_t)$ has the following form:

$$p(x_0|x_t) = \text{Normalize}\left(\exp\frac{-(x_0 - \mu)^2}{2\sigma^2} \, p(x_0)\right) \qquad \text{where } \mu = \frac{x_t}{c_0} \quad \sigma = \frac{c_1}{c_0} \qquad (13)$$

Here, $p(x_0)$ is the hidden data distribution, which is unknown and cannot be sampled directly. It can only be randomly selected from the existing samples $\{X_0^0, X_0^1, \dots, X_0^N\}$ ($X_0^i \sim p(x_0)$). The selection process can be considered as sampling from the following mixed Dirac delta distribution:$p(x_0) = \frac{1}{N}\sum_{i=0}^{N}\delta\left(x_0 - X_0^i\right)$. Substituting this into equation 13, we get:

$$p(x_0|x_t) = \frac{1}{Z_c}\exp\frac{-(x_0 - \mu)^2}{2\sigma^2}\sum_{i=0}^{N}\delta\left(x_0 - X_0^i\right) \qquad (14)$$

Here, $\mu = \frac{x_t}{c_0}$, $\sigma = \frac{c_1}{c_0}$, and $Z_c$ is normalization factor. It can be seen that when $p(x_0)$ is discrete, $p(x_0|x_t)$ is also discrete, and the probability of each discrete value $X_0^i$ is **inversely** proportional to **the distance between $X_0^i$ and $\mu$**. A similar conclusion is also presented in Appendix B of Karras et al. (2022), although the derivation method differ.

### 3.2 WEIGHTED SUM DEGRADATION PHENOMENON

We further analyze the characteristics of the mean of $p(x_0|x_t)$. According to the definition of expectation, the mean of $p(x_0|x_t)$ can be expressed as:

$$\int x_0 \, p(x_0|x_t)dx_0 = \frac{1}{Z_c}\sum_{i=0}^{N}X_0^i \, \exp\frac{-(X_0^i - \mu)^2}{2\sigma^2} \qquad (15)$$

The mean of $p(x_0|x_t)$ is a weighted sum of all $X_0^i$ samples, and the weight is inversely proportional to the distance between $X_0^i$ and $\mu$. If one sample is much closer than all others, the weighted sum degrades to that single sample. This is more likely with sparse data.

Figure 1 presents an example with sparse data ($X_0$, blue), and small noise std (green circle). In this case, most of $X_t$ remain near its origin data sample. This make $p(x_0|x_t)$ highly peaked at the closest $X_0$, causing its mean to degrade from a weighted sum to that single sample. We call this phenomenon **weighted sum degradation** and argue it potentially hinders the model learning the true data distribution.

Next, we analyze *weighted sum degradation* for conditional ImageNet-256 and ImageNet-512(Deng et al., 2009). Both datasets have high pixel dims (196608 and 786432) and retain high latent dims (4096 and 16480) after VAE(Kingma et al., 2013; Rombach et al., 2022) compression. As compression is typical, we will only consider the compressed case below.

We calculate the proportion of degradation. First, we sample $X_t$ as in training (first randomly select an $X_0$, then sample $X_t$ from $p(x_t|x_0 = X_0)$). Then, we determine whether $p(x_0|x_t = X_t)$ is degraded. If there exists an $X_0'$ such that $p(x_0 = X_0'|x_t = X_t) > 0.9$, then we consider *weighted sum degradation* to be present; if $X_0' = X_0$, it is called *weighted sum degradation to $X_0$*.

Since noise level also affects weighted sum degradation, we calculate degradation rates separately for different $t$. We calculate the proportions of both *weighted sum degradation* and *degradation to $X_0$* under two noise mixing schemes: VP (equation 1) and Flow Matching (equation 2).

Tables 1 and 2 present statistics for both datasets, showing several clear patterns:

Table 1: Statistics of ImageNet-256(weighted sum degradation / weighted sum degradation to $X_0$)

| merging\time | 200 | 300 | 400 | 500 | 600 | 700 | 800 | 900 |
|---|---|---|---|---|---|---|---|---|
| **vp** | 1.00/1.00 | 1.00/1.00 | 1.00/0.98 | 0.91/0.57 | 0.41/0.01 | 0.02/0.00 | 0.00/0.00 | 0.00/0.00 |
| **flow** | 1.00/1.00 | 1.00/1.00 | 1.00/1.00 | 1.00/1.00 | 1.00/0.95 | 0.97/0.69 | 0.76/0.15 | 0.09/0.00 |

Table 2: Statistics of ImageNet-512(weighted sum degradation / weighted sum degradation to $X_0$)

| merging\time | 200 | 300 | 400 | 500 | 600 | 700 | 800 | 900 |
|---|---|---|---|---|---|---|---|---|
| **vp** | 1.00/1.00 | 1.00/1.00 | 1.00/0.98 | 0.98/0.57 | 0.87/0.08 | 0.50/0.00 | 0.03/0.00 | 0.00/0.00 |
| **flow** | 1.00/1.00 | 1.00/1.00 | 1.00/1.00 | 1.00/1.00 | 1.00/0.94 | 0.99/0.67 | 0.95/0.20 | 0.71/0.01 |

- As $t$ decreases, the *weighted sum degradation* phenomenon becomes more pronounced.
- The degradation rate of Flow Matching is higher than that of VP.
- The higher the dimension, the greater the proportion of degradation.

Besides, we observe severe degradation in both datasets for both VP and Flow Matching, especially for $t < 600$. Furthermore, due to limited sampling during training, each $p(x_0|x_t = X_t)$ cannot be sufficiently sampled, so **the actual degradation ratio should be higher than the statistics show**.

In high dimensions, each $p(x_0|x_t = X_t)$ should be complex. When *weighted sum degradation* occurs, it is equivalent to using a single sample as an estimator of the mean, which typically have large error. If we cannot provide an accurate fitting target, we argue that the model is unlikely to learn the ideal target accurately. Therefore, **it is necessary to reconsider if diffusion models can truly learn the hidden probability distribution and how they work**.

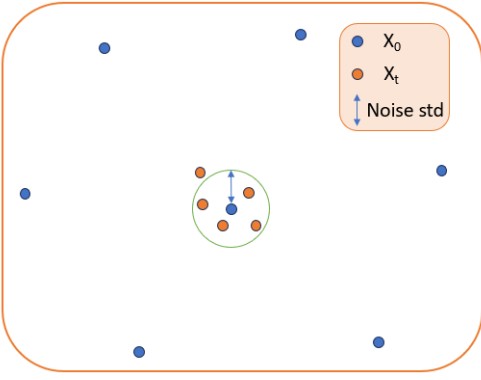

Figure 1: Impact of data sparsity on posterior probability distribution

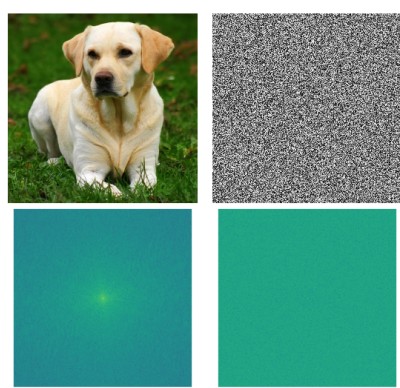

Figure 2: Left: Natural image and its spectrum. Right: noise and its spectrum

### 3.3 A SIMPLE WAY TO UNDERSTAND THE OBJECTIVE FUNCTION

As shown previously, weighted sum degradation is significant in high dimensions, which reduces the fitting target to the original data sample ($X_0$). Therefore, we can understand the objective in a simple way:**predict the original data sample ($X_0$) from the noise-mixed sample ($X_t$)** .

From the perspective of the frequency spectrum, we can further understand the principle (Dieleman, 2024).

As seen in Figure 2, natural image spectra concentrate energy in low frequencies (bright centrally, dark peripherally), while noise have a uniform spectrum. Thus, when mixed with noise, high frequencies always have lower SNR (signal-noise-ratio) than low frequencies. As noise grows, high frequencies are submerged first, then low frequencies (Figure 3).

When training a model to predict $X_0$ from noise-mixed samples, the model prioritizes frequencies based on their SNR. It easily predicts non-submerged frequencies (likely copying them).For submerged frequencies, it prioritizes predicting the lower-frequency components, as they have relatively higher SNR and larger amplitudes (giving them more weight in the Euclidean loss).

Thus, the objective can be further understood as **filtering higher-frequency components – completing the filtered frequency components** (Figure 4). At large $t$, even some low frequencies are submerged, so the model prioritizes predicting low frequencies. At small $t$, only high frequencies are submerged, and the model works on predicting these details. This frequency-dependent process is confirmed during inference: early steps (large $t$) generate contours, while later steps (small $t$) add details. Since the model compensates for the submerged frequencies, it can also be regarded as an **information enhancement operator**.

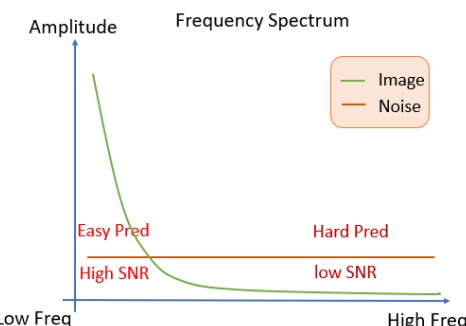

Figure 3: Image and noise frequency spectrum

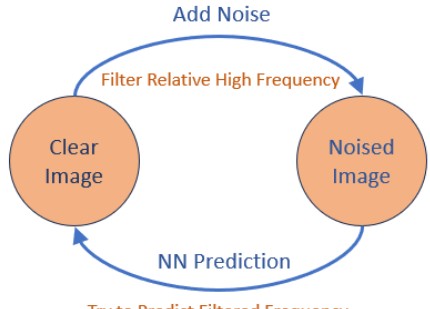

Figure 4: New perspective of training object function

## 4 A UNIFIED INFERENCE FRAMEWORK-NATURAL INFERENCE

We know that the inference methods of diffusion models rely on an assumption that the model can learn the hidden probability distributions or statistical quantities. However, as pointed out in the Section 3.2, in high-dimensional spaces the degradation phenomenon prevents the model from effectively learning these quantities. Therefore, it is necessary to attempt to reinterpret existing inference methods from a new perspective. Moreover, we have also seen in the Section 3.3 that the degraded objective function can be understood in a simple way - predicting the original image $x_0$ from a noisy image $x_t$. Based on the principle of **train-test matching**, this naturally leads us to ask: can current inference methods also be understood in a similar, simpler way?

The answer is yes. Below, we will reveal that most of inference methods can be unified into a simple framework based on predicting $x_0$, including Ancestor Sampling, DDIM, Euler, DPMsolver, DPMSolver++, DEIS, and Flow Matching solvers, among others.

We first introduce a class of key operations contained in the new framework.

### 4.1 SELF GUIDANCE

Following the concept of Classifier Free Guidance (Ho & Salimans, 2022), we introduce a new operation called Self Guidance. The principle of Classifier Free Guidance can be summarized as follows:

$$I_{out} = I_{bad} + \lambda \cdot (I_{good} - I_{bad}) \tag{16}$$

where $I_{bad}$ is the output of a less capable model, $I_{good}$ is the output of a more capable model, and both models share the same input. $\lambda$ controls the degree of guidance.

In fact, Classifier Free Guidance is somewhat similar to Unsharp Masking algorithm in traditional image enhancing processing(Gonzalez & Woods, 2017; scikit-image Development Team, 2013). In Unsharp Masking algorithm, $I_{good}$ is the original image, and $I_{bad}$ is the image after Gaussian

blur. The term $(I_{good} - I_{bad})$ provides the edge information, which, when added to the original image $I_{good}$, results in an image with sharper edges. Therefore, Classifier Free Guidance can also be considered as an **image enhancement** operation.

In the diffusion model inference process, a series of predicted $x_0$ are generated, where the quality of $x_0$ starts poor and improves over time. If an earlier predicted $x_0$ is used as $I_{bad}$ and a later predicted $x_0$ is used as $I_{good}$, then in this paper, we refer to this operation as Self Guidance, because both $I_{bad}$ and $I_{good}$ are outputs of the same model, and no additional model is needed.

Based on the value of $\lambda$, we further classify Self Guidance as follows:

- When $\lambda > 1$, it is called **Fore Self Guidance**, where the output improves the quality. See Fig. 6(c).
- When $0 < \lambda < 1$, it is called **Mid Self Guidance**, where the output is a linear interpolation between $I_{bad}$ and $I_{good}$, with a quality worse than $I_{good}$ but better than $I_{bad}$. See Fig. 6(d).
- When $\lambda < 0$, it is called **Back Self Guidance**, where the output is not only worse than $I_{good}$, but also worse than $I_{bad}$. See Fig. 6(e).

As shown in Appendix B, **the linear combination of any two model outputs can be viewed as a single Self Guidance, while the linear combination of multiple model outputs can be viewed as a composition of multiple Self Guidances**.

### 4.2 NATURAL INFERENCE

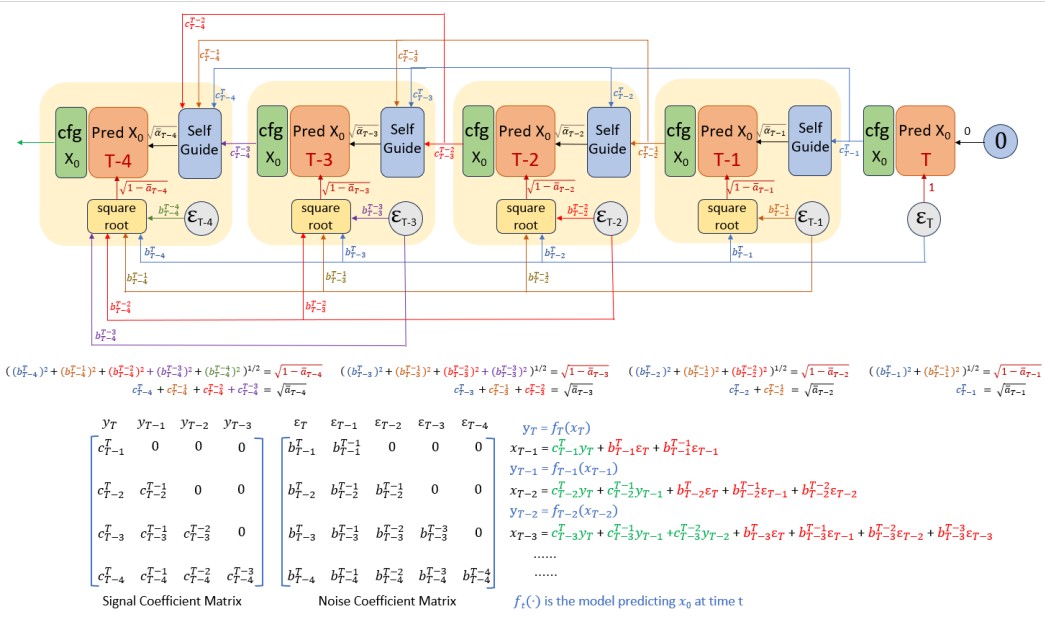

Figure 5: A new inference framework - Natural Inference

The new inference framework is illustrated in Figure 5, with the core ideas summarized as follows:

- It consists of $T$ models that predict $X_0$;
- Each model takes two part inputs: signal (image) and noise;
- The image signal is a linear combination of outputs from previous models, while the noise is a linear combination of previous noise and newly added noise;
- At time $t$, the sum of the coefficients corresponding to the image signal ($\sum_{i=t+1}^{T} c_t^i$) equals $\sqrt{\bar{\alpha}_t}$, and the square root of the sum of the squared noise coefficients ($\sqrt{\sum_{i=t}^{T} (b_t^i)^2}$) equals

$\sqrt{1 - \bar{\alpha}_t}$. This means that **the magnitudes of the signal and noise remain consistent with those used during the training phase**.

As shown in Section 4.1, **the linear combination of image signals can be interpreted as a composition of multiple Self Guidance operations**. The linear combination of independent noise is still noise (Taboga, 2021). Since the input signal of each model depends only on the output signals of previous models, the inference framework exhibits an **autoregressive** structure.

In this paper, we refer to $\sqrt{\bar{\alpha}_t}$ as the **marginal signal coefficient**, and $\sqrt{1 - \bar{\alpha}_t}$ as the **marginal noise coefficient**. The term $\sum_{i=t+1}^{T} c_t^i$ is referred to as the **equivalent marginal signal coefficient**, and $\sqrt{\sum_{i=t}^{T}(b_t^i)^2}$ is called the **equivalent marginal noise coefficient**. For clarity, all coefficients are organized into matrix form, as shown in the lower part of Figure 5. Due to the autoregressive property, the signal coefficient matrix has a lower triangular structure.

### 4.3 REPRESENT SAMPLING METHODS WITH NATURAL INFERENCE FRAMEWORK

This section briefly demonstrates how various sampling methods can be reformulated within the Natural Inference Framework. For more detailed explanations, please refer to Appendix C.

For first-order sampling methods (including DDPM, DDIM, ODE Euler, SDE Euler, and Flow Matching ODE Euler), their iterative procedures can all be expressed in the following form:

$$y_t = f_t(x_t) \tag{17}$$

$$x_{t-1} = d_{t-1} \cdot x_t + e_{t-1} \cdot y_t + g_{t-1} \cdot \epsilon_{t-1} \tag{18}$$

Here, $f_t$ is the model function predicting $x_0$ at step $t$. $x_t$, $y_t$, and $\epsilon_{t-1}$ are vectors, while $d_{t-1}$, $e_{t-1}$, and $g_{t-1}$ are fixed scalars. For deterministic methods, $g_{t-1}$ is zero.

Starting from $x_T$, we can iterate according to the above equation to further determine the expressions of $x_{T-1}, x_{T-2}, \cdots, x_1$, and $x_0$. Each $x_t$ can be represented as two components: one is a linear combination of $\{y_i\}_{i=t+1}^{T}$, and the other is a linear combination of $\{\varepsilon_i\}_{i=t}^{T}$. Since $d_{t-1}$, $e_{t-1}$, and $g_{t-1}$ are all known constants, the weights for each element in $\{y_i\}_{i=t+1}^{T}$ and $\{\varepsilon_i\}_{i=t}^{T}$ can be calculated.

The calculation results show that the sum of the coefficients corresponding to $\{y_i\}_{i=t+1}^{T}$ is approximately equal to $\sqrt{\bar{\alpha}_t}$, and the square root of the sum of squared coefficients for $\{\varepsilon_i\}_{i=t}^{T}$ is approximately $\sqrt{1 - \bar{\alpha}_t}$. Moreover, the approximation error decreases as the number of sampling steps increases (see Figures 7-9 and Figures 13-14). Therefore, these sampling methods can be represented in the form of the Natural Inference framework.

The above computation can be quite complex, especially when the number of sampling steps is large. Therefore, it is necessary to seek more efficient computation methods. Symbolic computation software (Team, 2013) offers a promising solution. With minor modifications to the original algorithm code, it can automatically compute the expression for each $x_t$. For more detailed information, please refer to the accompanying code.

For higher-order sampling methods, their iteration rules are relatively complex, but the expression for $x_t$ can also be quickly calculated with the help of symbolic computation software. The calculation results indicate that DPMSOLVER, DPMSOLVER++, and DEIS yield results similar to those of first-order sampling methods (see Figures 10-12).

Appendix C.6 also provides a simple and intuitive example that represent the five-step Euler Inference method with the form of Natural Inference.

### 4.4 ADVANTAGES OF THE NATURAL INFERENCE FRAMEWORK

Thus, we have used a completely new perspective to explain high-dimensional diffusion models, including the objective function during training and the inference algorithm during testing. This new perspective has several advantages:

- The new perspective maintains **training-testing consistency**, where the goal during training is to predict $x_0$, and the goal during testing is also to predict $x_0$.

- The new perspective divides the inference process into a series of operations for predicting $x_0$, each of which has clear input image signals and output image signals. This makes the inference process **more visual and interpretable**, providing significant help for debugging and problem analysis. Figures 15 and 16 provide a visualization of the complete inference process.

- As discussed in Section 3.3, predicting $x_0$ can be regarded as an information enhancement operator. Similarly, Section 4.1 shows that classifier-free guidance can also be viewed as an information enhancement operator. Therefore, the entire inference process can be understood as a progressive enhancement of information, a process that does not require any statistical knowledge.

- From this new perspective, existing sampling algorithms are merely specific parameter configurations within the Natural Inference framework. Within this framework, other, potentially more optimal parameter configurations may exist that can generate higher-quality samples. Exploring these possibilities could be a direction for future work.

## 5 CONCLUSION

This paper investigates the operational principles of high-dimensional diffusion models. We first analyze the objective function and explore the impact of data sparsity in high-dimensional settings, demonstrating that, due to such sparsity, these models cannot effectively learn the underlying probability distributions or their key statistical quantities. Building on this insight, we propose a novel perspective for interpreting the objective function. In addition, we introduce a new inference framework that not only unifies most inference methods but also aligns with the degraded objective function. This framework offers an intuitive understanding of the inference process without relying on any statistical concepts. We hope that this work will encourage the community to rethink the operational principles of high-dimensional diffusion models and further enhance their training and inference methodologies.

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

## A  ADDITIONAL PROOFS

### A.1  PREDICTING POSTERIOR MEAN IS EQUIVALENT TO PREDICTING $X_0$

In the following, we will prove that the following two objective functions are equivalent:

$$\min_\theta \int p(x_t) \left\| f_\theta(x_t) - \int p(x_0|x_t)\, x_0\, dx_0 \right\|^2 dx_t \iff \min_\theta \iint p(x_0, x_t) \left\| f_\theta(x_t) - x_0 \right\|^2 dx_0 dx_t$$

where, $f_\theta(x_t)$ is a neural network model.

Proof:

For $\left\| f_\theta(x_t) - \int p(x_0|x_t) x_0 dx_0 \right\|^2$, the following relation holds:

$$\left\| f_\theta(x_t) - \int p(x_0|x_t) x_0 dx_0 \right\|^2 \tag{19}$$

$$= \|f_\theta(x_t)\|^2 - 2 f_\theta(x_t) \int p(x_0|x_t) x_0 dx_0 + \left\| \int p(x_0|x_t) x_0 dx_0 \right\|^2 \tag{20}$$

$$= \int p(x_0|x_t) \|f_\theta(x_t)\|^2 dx_0 - 2 f_\theta(x_t) \int p(x_0|x_t) x_0 dx_0 + C_1 \tag{21}$$

$$= \int p(x_0|x_t) \left( \|f_\theta(x_t)\|^2 - 2 f_\theta(x_t) x_0 + x_0^2 \right) dx_0 - \int p(x_0|x_t) x_0^2 dx_0 + C_1 \tag{22}$$

$$= \int p(x_0|x_t) \left\| f_\theta(x_t) - x_0 \right\|^2 dx_0 - C_2 + C_1 \tag{23}$$

Where $C_1$ and $C_2$ are constants that do not depend on $\theta$. In equation 21, we apply $\|f_\theta(x_t)\|^2 = \|f_\theta(x_t)\|^2 \int p(x_0|x_t) dx_0 = \int p(x_0|x_t) \|f_\theta(x_t)\|^2 dx_0$.

Substituting the above relation into the objective function for predicting posterior mean, we get:

$$\int p(x_t) \left\| f_\theta(x_t) - \int p(x_0|x_t) x_0 dx_0 \right\|^2 dx_t \tag{24}$$

$$= \int p(x_t) \left( \int p(x_0|x_t) \left\| f_\theta(x_t) - x_0 \right\|^2 dx_0 - C_2 + C_1 \right) dx_t \tag{25}$$

$$= \iint p(x_0, x_t) \left\| f_\theta(x_t) - x_0 \right\|^2 dx_0 dx_t + \int p(x_t) \left( C_1 - C_2 \right) dx_t \tag{26}$$

$$= \iint p(x_0, x_t) \left\| f_\theta(x_t) - x_0 \right\|^2 dx_0 dx_t + C_3 \tag{27}$$

That is, the two objective functions differ only by a constant that does not depend on the optimization parameters. Therefore, the two objective functions are equivalent.

### A.2  CONDITIONAL SCORE

Below is the proof of the following relation:

$$\frac{\partial \log p(x_t)}{\partial x_t} = \int p(x_0|x_t) \frac{\partial \log p(x_t|x_0)}{\partial x_t} dx_0 \tag{28}$$

Proof:

$$\frac{\partial \log p(x_t)}{\partial x_t} = \frac{1}{p(x_t)} \frac{\partial p(x_t)}{\partial x_t} \tag{29}$$

$$= \frac{1}{p(x_t)} \frac{\partial \left( \int p(x_0) p(x_t|x_0) dx_0 \right)}{\partial x_t} \tag{30}$$

$$= \int \frac{p(x_0)}{p(x_t)} \frac{\partial p(x_t|x_0)}{\partial x_t} dx_0 \tag{31}$$

$$= \int \frac{p(x_0, x_t)/p(x_t)}{p(x_0, x_t)/p(x_0)} \frac{\partial p(x_t|x_0)}{\partial x_t} dx_0 \tag{32}$$

$$= \int \frac{p(x_0|x_t)}{p(x_t|x_0)} \frac{\partial p(x_t|x_0)}{\partial x_t} dx_0 \tag{33}$$

$$= \int p(x_0|x_t) \frac{\partial \log p(x_t|x_0)}{\partial x_t} dx_0 \tag{34}$$

### A.3 FORM OF THE POSTERIOR PROBABILITY

The following derivation is based on Zheng (2023) and Zheng (2024).

Assume that $x_t$ has the following form:

$$x_t = c_0 \cdot x_0 + c_1 \cdot \epsilon \qquad \text{where } c_0 \text{ and } c_1 \text{ is constant} \tag{35}$$

Then we have:

$$p(x_t|x_0) \sim \mathcal{N}(x_t; c_0 x_0, c_1^2) \tag{36}$$

According to Bayes' theorem, we have

$$p(x_0|x_t) = \frac{p(x_t|x_0)p(x_0)}{p(x_t)} \tag{37}$$

$$= \frac{p(x_t|x_0)p(x_0)}{\int p(x_t|x_0)p(x_0)dx_0} \tag{38}$$

$$= \text{Normalize}\big(p(x_t|x_0)p(x_0)\big) \tag{39}$$

where *Normalize* represents the normalization operator, and the normalization divisor is $\int p(x_t|x_0)p(x_0)dx_0$.

Substituting Equation equation 36 into this, we get:

$$p(x_0|x_t) = \text{Normalize}\left( \frac{1}{\sqrt{2\pi c_1^2}} \exp \frac{-(x_t - c_0 x_0)^2}{2c_1^2} \, p(x_0) \right) \tag{40}$$

$$= \text{Normalize}\left( \frac{1}{\sqrt{2\pi c_1^2}} \exp \frac{-(x_0 - \frac{x_t}{c_0})^2}{2\frac{c_1^2}{c_0^2}} \, p(x_0) \right) \tag{41}$$

$$= \text{Normalize}\left( \exp \frac{-(x_0 - \mu)^2}{2\sigma^2} \, p(x_0) \right) \tag{42}$$

$$\text{where } \mu = \frac{x_t}{c_0} \qquad \sigma = \frac{c_1}{c_0} \tag{43}$$

In the above derivation, due to the presence of the normalization operator, we can ignore the factor $\frac{1}{\sqrt{2\pi c_1^2}}$.

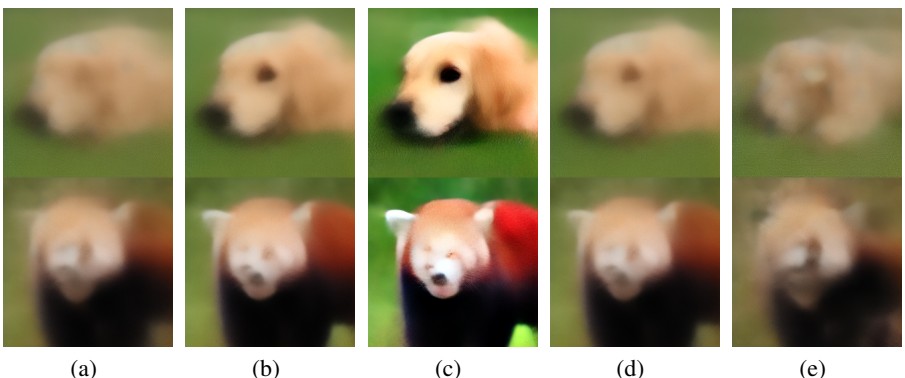

Figure 6: (a) Model output on t=540 ($I_{bad}$)    (b) Model output on t=500 ($I_{good}$)    (c) Output of Fore Self Guidence    (d) Output of Mid Self Guidence    (e) Output of Back Self Guidence

## B   SELF GUIDANCE AND ITS COMPOSITION

In this section, we show that the linear combination of any two model outputs(prediting $x_0$) can be viewed as a Self Guidance operation.

As described in Section 3.2, the self guidance is defined as follows:

$$I_{out} = I_{bad} + \lambda \cdot (I_{good} - I_{bad}) \tag{44}$$

This equation can be further written as

$$I_{out} = \lambda \cdot I_{good} + (1 - \lambda) \cdot I_{bad} \tag{45}$$

$$= \eta_{good} \cdot I_{good} + \eta_{bad} \cdot I_{bad} \tag{46}$$

$$\text{where} \quad \eta_{good}, \eta_{bad} \in real \quad \eta_{good} + \eta_{bad} = 1 \tag{47}$$

As shown above, the coefficients of $I_{bad}$ and $I_{good}$ can take any value, but the sum of $I_{bad}$ and $I_{good}$ must equal 1. For **Fore Self Guidance**, $\eta_{good} > 0$, $\eta_{bad} < 0$; for **Mid Self Guidance**, $\eta_{good} > 0$, $\eta_{bad} > 0$; for **Back Self Guidance**, $\eta_{good} < 0$, $\eta_{bad} > 0$.

For the linear combination of any two model outputs, it can be written as:

$$I_{out} = a \cdot I_{good} + b \cdot I_{bad} = (a + b) \cdot (\frac{a}{a + b} \cdot I_{good} + \frac{b}{a + b} \cdot I_{bad}) \tag{48}$$

Since the sum of the two coefficients equals 1, the operation inside the parentheses is a Self Guidance operation.

Thus, **the linear combination of any two $I_{bad}$ and $I_{good}$ can be represented as Self Guidance with a scaling factor**.

For the linear combination of multiple model outputs, it can be written as:

$$I_{out} = a \cdot I_a + b \cdot I_b + c \cdot I_c \tag{49}$$

$$= (a + b) \cdot (\frac{a}{a + b} \cdot I_a + \frac{b}{a + b} \cdot I_b) + c \cdot I_c \tag{50}$$

$$= (a + b + c) \cdot (\frac{a + b}{a + b + c} \cdot (\frac{a}{a + b} I_a + \frac{b}{a + b} I_b) + \frac{c}{a + b + c} \cdot I_c) \tag{51}$$

Thus, **the linear combination of multiple model outputs can be viewed as a composition of Self Guidences**.

# C  REPRESENT SAMPLING METHODS WITH NATURAL INFERENCE FRAMEWORK

## C.1  REPRESENT DDPM ANCESTRAL SAMPLING WITH NATURAL INFERENCE FRAMEWORK

This subsection will demonstrate that the DDPM Ancestor Sampling can be reformulated within the Natural Inference framework. The iterative process of the Ancestor Sampling is as follows:

$$y_t = f_t(x_t)$$
$$x_{t-1} = d_{t-1} \cdot x_t + e_{t-1} \cdot y_t + g_{t-1} \cdot \epsilon_{t-1} \tag{52}$$
$$\text{where } d_{t-1} = \frac{\sqrt{\alpha_t}(1 - \bar{\alpha}_{t-1})}{1 - \bar{\alpha}_t} \quad e_{t-1} = \frac{\sqrt{\bar{\alpha}_{t-1}}\beta_t}{1 - \bar{\alpha}_t} \quad g_{t-1} = \sqrt{\frac{1 - \bar{\alpha}_{t-1}}{1 - \bar{\alpha}_t}\beta_t}$$

Here, $f_t$ is the model function at the $t$-th step. In this case, we assume the model predicts $x_0$, but other forms of prediction models (such as predict $\varepsilon$ or predict $v$) can be transformed into the form of predicting $x_0$. $y_t$ is the output of $f_t$, which is the predicted $x_0$ at the $t$-th step.

According to the above iterative algorithm, $x_{T-1}$ can be expressed as

$$x_T = g_T \cdot \epsilon_T \quad \text{where } g_T = 1$$
$$x_{T-1} = d_{T-1} \cdot x_T + e_{T-1} \cdot y_T + g_{T-1} \cdot \epsilon_{T-1} \tag{53}$$
$$= e_{T-1}y_T + (d_{T-1}g_T\epsilon_T + g_{T-1}\epsilon_{T-1})$$

Based on the expression of $x_{T-1}$, the expression of $x_{T-2}$ can be written as

$$x_{T-2} = d_{T-2} \cdot x_{T-1} + e_{T-2} \cdot y_{T-1} + g_{T-2} \cdot \epsilon_{T-2}$$
$$= (d_{T-2}e_{T-1}y_T + e_{T-2}y_{T-1}) + (d_{T-2}d_{T-1}g_T\epsilon_T + d_{T-2}g_{T-1}\epsilon_{T-1} + g_{T-2}\epsilon_{T-2}) \tag{54}$$

Based on the expression of $x_{T-2}$, the expression of $x_{T-3}$ can be further written as

$$x_{T-3} = d_{T-3} \cdot x_{T-2} + e_{T-3} \cdot y_{T-2} + g_{T-3} \cdot \epsilon_{T-3}$$
$$= (d_{T-3}d_{T-2}e_{T-1}y_T + d_{T-3}e_{T-2}y_{T-1} + e_{T-3}y_{T-2})$$
$$+ (d_{T-3}d_{T-2}d_{T-1}g_T\epsilon_T + d_{T-3}d_{T-2}g_{T-1}\epsilon_{T-1} + d_{T-3}g_{T-2}\epsilon_{T-2} + g_{T-3}\epsilon_{T-3}) \tag{55}$$

Similarly, each $x_t$ can be recursively written in a similar form. It can be observed that each $x_t$ can be decomposed into two parts: one part is a weighted sum of past predictions of $x_0$ (i.e., $y_t$), and the other part is a weighted sum of past noise and newly added noise. Since $d_t$, $e_t$, and $g_t$ are all known constants, the equivalent signal coefficient and equivalent noise coefficient for each $x_t$ can be accurately computed.

The computation results show that the equivalent signal coefficient of each $x_t$ is almost equal to $\sqrt{\bar{\alpha}_t}$, and the equivalent noise coefficient is approximately $\sqrt{1 - \bar{\alpha}_t}$. Moreover, the slight error diminishes as the number of sampling steps $T$ increases. Specifically, Figure 7 illustrates the results for 18 steps, 100 steps, and 500 steps.

Table 3 presents the complete coefficients of each $x_t$ with respect to $y_t$ in matrix form, where each row corresponds to an $x_t$. Table 4 provides the complete coefficients of each $x_t$ with respect to $\epsilon_t$ in matrix form. It can be seen that the noise coefficient matrix differs slightly from the signal coefficient matrix, with an additional nonzero coefficient appearing to the right of the diagonal elements. This indicates that a small amount of new noise is introduced at each step, causing the overall noise *pattern* to change at a slow rate.

At this point, we have successfully demonstrated that the DDPM Ancestral Sampling process can be represented using the Natural Inference framework.

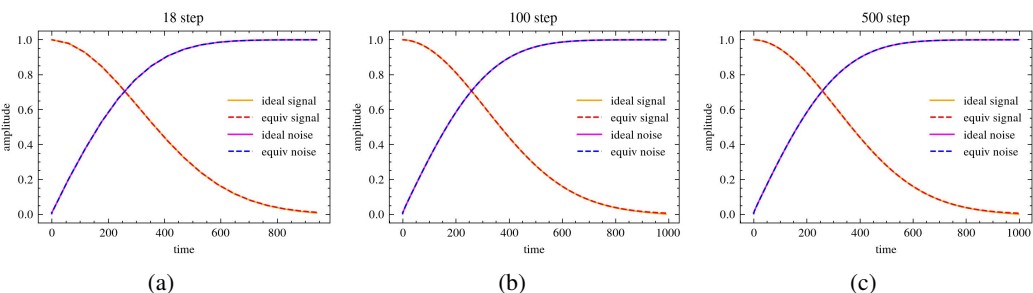

Figure 7: DDPM equivalent marginal coefficients and ideal margingal coefficients    (a) 18 step    (b) 100 step    (c) 500 step

## C.2 REPRESENT DDIM WITH NATURAL INFERENCE FRAMEWORK

The iterative rule of the DDIM can be expressed in the following form:

$$
\begin{aligned}
y_t &= f_t(x_t) \\
x_{t-1} &= \sqrt{\bar{\alpha}_{t-1}} \cdot x_t + \sqrt{1 - \bar{\alpha}_{t-1}} \cdot \frac{x_t - \sqrt{\bar{\alpha}_t} y_t}{\sqrt{1 - \bar{\alpha}_t}} \cdot y_t \\
&= d_{t-1} \cdot x_t + e_{t-1} \cdot y_t \\
\text{where } d_{t-1} &= \frac{\sqrt{1 - \bar{\alpha}_{t-1}}}{\sqrt{1 - \bar{\alpha}_t}} \quad e_{t-1} = \left(\sqrt{\bar{\alpha}_{t-1}} - \frac{\sqrt{1 - \bar{\alpha}_{t-1}}}{\sqrt{1 - \bar{\alpha}_t}\sqrt{\bar{\alpha}_t}}\right)
\end{aligned}
\tag{56}
$$

It can be seen that the iterative rule of DDIM is similar to those of DDPM Ancestral Sampling, except that the term $g_{t-1} \cdot \epsilon_{t-1}$ is missing, meaning that no new noise is added at each step. Following the recursive way of DDPM Ancestral Sampling, each $x_t$ corresponding to $t$ can also be written in a similar form, as follows:

$$
\begin{aligned}
x_T &= g_T \cdot \epsilon_T \qquad \text{where } g_T = 1 \\
x_{T-1} &= e_{T-1} y_T + d_{T-1} g_T \epsilon_T \\
x_{T-2} &= (d_{T-2} e_{T-1} y_T + e_{T-2} y_{T-1}) + d_{T-2} d_{T-1} g_T \epsilon_T \\
x_{T-3} &= (d_{T-3} d_{T-2} e_{T-1} y_T + d_{T-3} e_{T-2} y_{T-1} + e_{T-3} y_{T-2}) \\
&\quad + d_{T-3} d_{T-2} d_{T-1} g_T \epsilon_T
\end{aligned}
\tag{57}
$$

It can be seen that the form of DDIM is slightly different from DDPM. Since DDIM does not introduce new noise at each step, there is only one noise term.

The computation results show that the equivalent signal coefficients of each $x_t$ are approximately equal to $\sqrt{\bar{\alpha}_t}$, and the equivalent noise coefficient contains only the term related to $\epsilon_T$, whose coefficient is almost equal to $\sqrt{1 - \bar{\alpha}_t}$. Figure 8 illustrates the results for 18 steps, 100 steps, and 500 steps, respectively. It can be observed that the errors in the equivalent coefficients are minimal and almost indistinguishable. Table 5 presents the complete signal coefficient matrix for 18 steps.

Therefore, the sampling process of DDIM can also be represented using the Natural Inference framework.

## C.3 REPRESENT FLOW MATCHING EULER SAMPLING WITH NATURAL INFERENCE FRAMEWORK

The noise mixing method of Flow Matching is shown in Equation equation 2. When using Euler discretized integral sampling, its iterative rule can be expressed as follows:

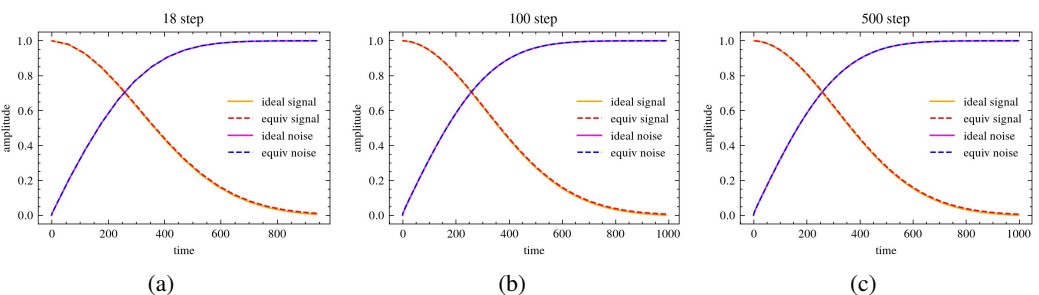

Figure 8: DDIM equivalent marginal coefficients and ideal marginal coefficients    (a) 18 step    (b) 100 step    (c) 500 step

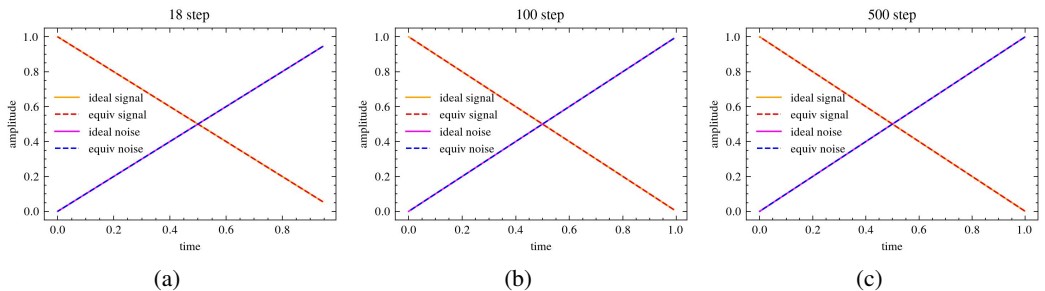

Figure 9: Flow matching euler sampler equivalent marginal coefficients and ideal marginal coefficients (a) 18 step    (b) 100 step    (c) 500 step

$$
\begin{aligned}
y_i &= f_i(x_i) \\
x_{i-1} &= x_i + (t_{i-1} - t_i)(-y_i + \epsilon) \\
&= x_i + (t_{i-1} - t_i)\frac{x_i - y_i}{t_i} \\
&= d_{i-1} \cdot x_i + e_{i-1} \cdot y_i \\
\text{where } d_{i-1} &= \frac{t_{i-1}}{t_{t_i}} \quad e_{i-1} = (1 - \frac{t_{i-1}}{t_{t_i}})
\end{aligned}
\tag{58}
$$

where $f_i$ is the model predicting $x_0$, and $y_i$ is the output of the model $f_i$ corresponding to the discrete time point $t_i$.

It can be observed that the iterative rule of the Euler algorithm in Flow Matching is similar to that of DDIM, so each $x_i$ can also be expressed in a similar form.

The computation results show that for each discrete point $t_i$, the equivalent signal coefficient of $x_i$ is **exactly equal to** $1 - t_i$, and the equivalent noise coefficient has only the $\epsilon_N$ term, whose coefficient is **exactly equal to** $t_i$. The specific results can be seen in Figure 9, which shows the results for 18 steps, 200 steps, and 500 steps, respectively. Table 6 presents the signal coefficient matrix for 18 steps.

### C.4    Represent high order samplers with Natural Inference framework

In the previous sections, most first-order sampling algorithms have already been represented using the Natural Inference framework. For second-order and higher-order sampling algorithms, since the update rules of $x_i$ are more complex, it is quite difficult to directly compute the expression of each $x_i$. Therefore, alternative solutions must be sought. Symbolic computation tools provide a suitable solution to this challenge, as they can automatically analyze complex mathematical expressions. With

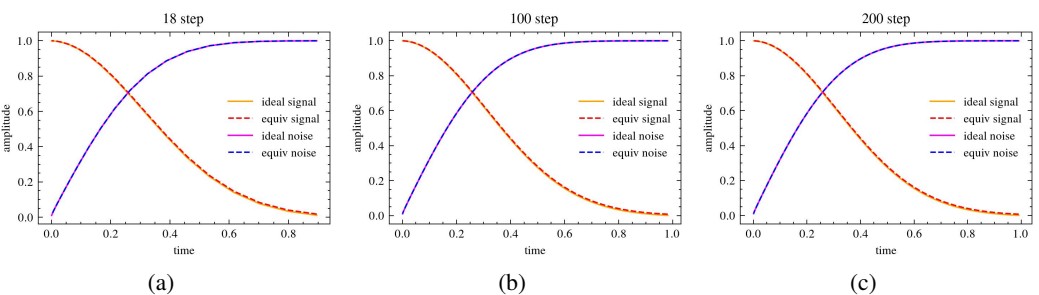

Figure 10: DEIS equivalent marginal coefficients and ideal marginal coefficients    (a) 18 step    (b) 100 step    (c) 500 step

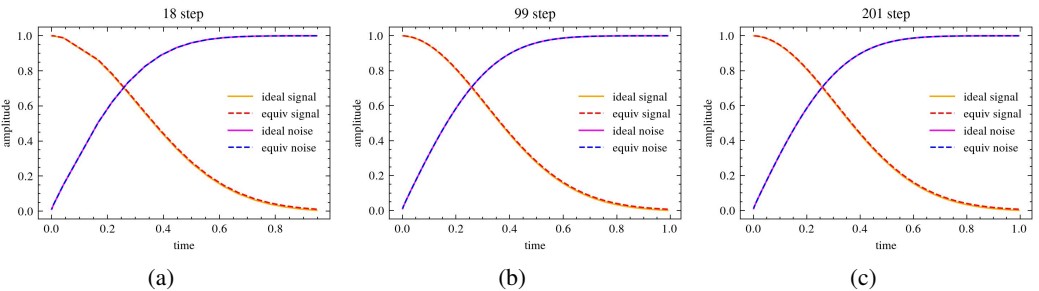

Figure 11: dpmsolver3s equivalent marginal coefficients and ideal marginal coefficients    (a) 18 step (b) 99 step    (c) 201 step

slight modifications to the original algorithm code, they can automatically compute the coefficients of each $y_i$ term and $\epsilon_i$ term. The toolkit used in this paper is SymPyTeam (2013), and For specific details, please refer to the code attached in this paper.

The compuation results show that DEIS, DPMSolver, and DPMSolver++ yield the same conclusion as DDIM: each $x_i$ can be decomposed into two parts, with its equivalent signal coefficient approximately equal to $\sqrt{\bar{\alpha}_i}$ and its equivalent noise coefficient approximately equal to $\sqrt{1 - \bar{\alpha}_i}$.

Figure 10 shows the results for the DEIS(tab3) algorithm, Figure 11 presents the results for the third-order DPMSolver, and Figure 12 illustrates the results for the second-order DPMSolver++. It can be observed that these higher-order sampling algorithms exhibit the same properties and can also be represented using the Natural Inference framework.

Table 7 provides the coefficient matrix for the third-order DEIS algorithm (18 steps). Table 8 and Table 9 present the coefficient matrices for the second-order and third-order DPMSolver algorithms (18 steps). Table 10 and Table 11 provide the coefficient matrices for the second-order and third-order DPMSolver++ algorithms (18 steps).

## C.5    REPRESENT SDE EULER AND ODE EULER WITH NATURAL INFERENCE FRAMEWORK

For SDE Euler and ODE Euler, the expressions for each $x_t$ can also be automatically computed using SymPy. The compuation results indicate that these two algorithms yield results similar to previous algorithms, but they suffer from relatively larger errors, especially when the number of steps is small. For details, see Figure 13 and Figure 14.

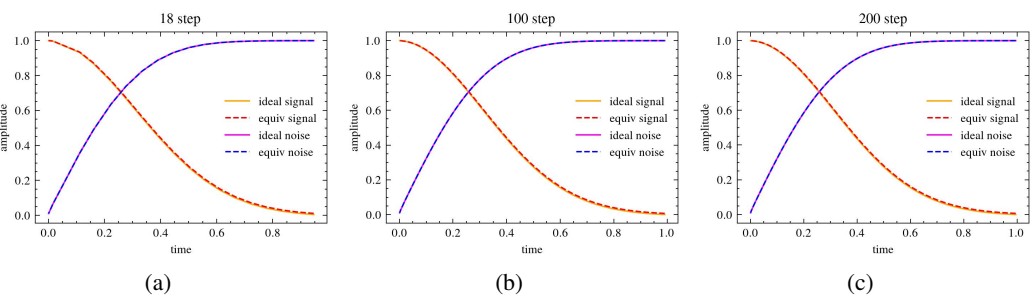

Figure 12: dpmsolver++2s equivalent marginal coefficients and ideal marginal coefficients    (a) 18 step    (b) 99 step    (c) 201 step

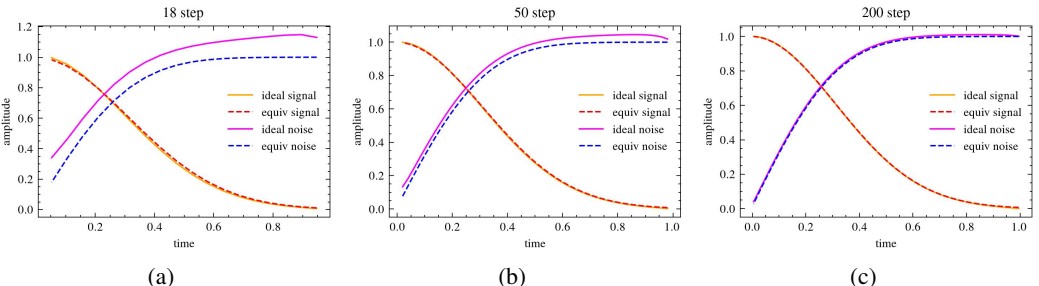

Figure 13: SDE Euler equivalent marginal coefficients and ideal marginal coefficients    (a) 18 step (b) 50 step    (c) 200 step

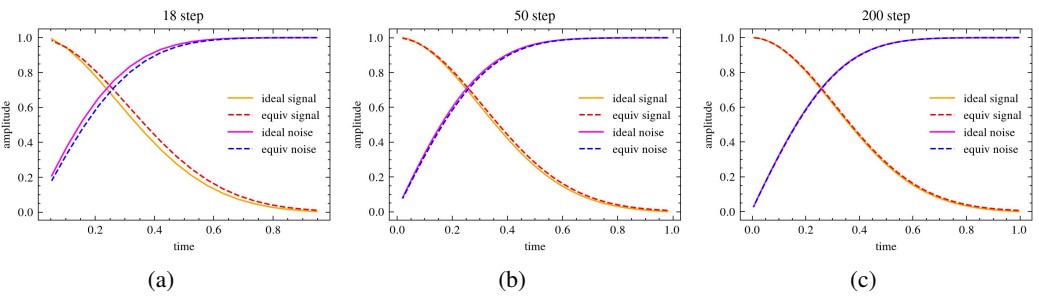

Figure 14: ODE Euler equivalent marginal coefficients and ideal marginal coefficients    (a) 18 step (b) 50 step    (c) 200 step

### C.6 A TOY EXAMPLE - REPRESENT FLOW MATCHING'S EULER SAMPLING WITH NATURAL INFERENCE

For the inference method of Flow Matching, it is equivalent to solving the following ODE:

$$\frac{d\mathbf{x}_t}{dt} = \mu_\theta(\mathbf{x}_t, t), \qquad \mathbf{x}_t = (1-t) \cdot \mathbf{x}_0 + t \cdot \varepsilon. \tag{59}$$

Here, $\mu_\theta(\mathbf{x}_t, t)$ is a neural network model, which is trained to fit the degenerated target $(\varepsilon - \mathbf{x}_0)$.

Considering that $\mathbf{x}_t$ can be written in the form of Eq. (2), and that $\mu_\theta(\mathbf{x}_t, t)$ is a model predicting $(\varepsilon - \mathbf{x}_0)$, we have

$$\mathbf{x}_t = (1-t) \cdot \mathbf{x}_0 + t \cdot \varepsilon = \mathbf{x}_0 + t \cdot (\varepsilon - \mathbf{x}_0). \tag{60}$$

Then, $\mathbf{x}_t - t \cdot \mu_\theta(\mathbf{x}_t, t)$ is a function that predicts $\mathbf{x}_0$, denoted as $f_\theta^{\mathbf{x}_0}(\mathbf{x}_t, t)$, i.e.,

$$f_\theta^{\mathbf{x}_0}(\mathbf{x}_t, t) = \mathbf{x}_t - t \cdot \mu_\theta(\mathbf{x}_t, t). \tag{61}$$

It can be observed that $f_\theta^{\mathbf{x}_0}(\mathbf{x}_t, t)$ is also a function of $t$ and $\mathbf{x}_t$. For convenience of notation, we abbreviate $f_\theta^{\mathbf{x}_0}(\mathbf{x}_t, t)$ as $f_t^{\mathbf{x}_0}$.

#### C.6.1 A SIMPLE EULER INFERENCE EXAMPLE

The Euler method is the most basic approach for solving ODEs. Below we illustrate it with a 5-step example. We discretize the continuous interval $t \in [1, 0]$ into 6 time points: $[1, 0.8, 0.6, 0.4, 0.2, 0.0]$.

First, we randomly initialize a noise sample (denoted as $\varepsilon_s$) as the starting point of integration(i.e. $\hat{x}_{1.0} = \varepsilon_s$). We substitute $t = 1.0$ and $x_t = \hat{x}_{1.0}$ into the model function $\mu_\theta(x_t, t)$, run the model, and obtain the **velocity field** at $t = 1.0$, denoted by $\mu_\theta(\hat{x}_{1.0}, 1.0)$.

We then perform **the first** update:

$$\hat{\mathbf{x}}_{0.8} = \hat{\mathbf{x}}_{1.0} + (0.8 - 1.0) \cdot \mu_\theta(\hat{\mathbf{x}}_{1.0}, 1.0) \tag{62}$$

$$= \hat{\mathbf{x}}_{1.0} - 0.2 \cdot \frac{\hat{\mathbf{x}}_{1.0} - f_{1.0}^{\mathbf{x}_0}}{1.0} \tag{63}$$

$$= 0.8 \cdot \varepsilon_s + 0.2 \cdot f_{1.0}^{\mathbf{x}_0}. \tag{64}$$

From equation 64, we can see that $\hat{x}_{0.8}$ consists of two components: the **initial random noise** and the **model output predicting** $x_0$, with coefficients 0.8 and 0.2, respectively. Moreover, we know that during training, the constructed $x_{0.8}$ satisfies

$$x_{0.8} = 0.8 \cdot \varepsilon + 0.2 \cdot x_0. \tag{65}$$

Comparing the two expressions, we observe that $\hat{x}_{0.8}$ **and** $x_{0.8}$ **have very similar structures: both are composed of noise and signal with identical proportions; the only difference is that** *predicted* $x_0$ **replaces** *original* $x_0$**.**

We now perform **the second** update:

Substitute $t = 0.8$ and $\mathbf{x}_t = \hat{\mathbf{x}}_{0.8}$ into the model function $\mu_\theta(\mathbf{x}_t, t)$, run the model, and obtain the velocity field at $t = 0.8$, denoted as $\mu_\theta(\hat{\mathbf{x}}_{0.8}, 0.8)$.

$$\hat{\mathbf{x}}_{0.6} = \hat{\mathbf{x}}_{0.8} + (0.6 - 0.8) \cdot \mu_\theta(\hat{\mathbf{x}}_{0.8}, 0.8) \tag{66}$$

$$= \hat{\mathbf{x}}_{0.8} - 0.2 \cdot \frac{\hat{\mathbf{x}}_{0.8} - f_{0.8}^{\mathbf{x}_0}}{0.8} \tag{67}$$

$$= 0.75 \cdot \hat{\mathbf{x}}_{0.8} + 0.25 \cdot f_{0.8}^{\mathbf{x}_0} \tag{68}$$

Replacing $\hat{\mathbf{x}}_{0.8}$ in equation 68 using equation 64, we obtain:

$$\hat{\mathbf{x}}_{0.6} = 0.75 \cdot \hat{\mathbf{x}}_{0.8} + 0.25 \cdot f_{0.8}^{\mathbf{x}_0} \tag{69}$$

$$= 0.75 \cdot (0.8 \cdot \varepsilon_s + 0.2 \cdot f_{1.0}^{\mathbf{x}_0}) + 0.25 \cdot f_{0.8}^{\mathbf{x}_0} \tag{70}$$

$$= 0.6 \cdot \varepsilon_s + 0.4 \cdot \left( \frac{0.15}{0.4} \cdot f_{1.0}^{\mathbf{x}_0} + \frac{0.25}{0.4} \cdot f_{0.8}^{\mathbf{x}_0} \right) \tag{71}$$

The expression inside the parentheses in equation 71 can be viewed as a **linear interpolation of** $f_{1.0}^{\hat{x}_0}$ **and** $f_{0.8}^{\hat{x}_0}$**, which can also be interpreted as a form of Self Guidance.**

Again, we observe that **the structure of** $\hat{x}_{0.6}$ **is identical to that of** $x_{0.6}$ **constructed during training: both consist of noise and signal with the same proportions; the only difference is that** $x_{0.6}$ **uses the** *original* $x_0$**, while** $\hat{x}_{0.6}$ **uses a linear weighted sum of** *previous predicted* $x_0$ **as its signal component.**

We now proceed to **the third** update:

$$\hat{\mathbf{x}}_{0.4} = \hat{\mathbf{x}}_{0.6} + (0.4 - 0.6) \cdot \mu_\theta(\hat{\mathbf{x}}_{0.6}, 0.6) \tag{72}$$

$$= \hat{\mathbf{x}}_{0.6} - 0.2 \cdot \frac{\hat{\mathbf{x}}_{0.6} - f_{0.6}^{\mathbf{x}_0}}{0.6} \tag{73}$$

$$= \frac{2}{3} \cdot \hat{\mathbf{x}}_{0.6} + \frac{1}{3} \cdot f_{0.6}^{\mathbf{x}_0} \tag{74}$$

$$= \frac{2}{3} \left( 0.6 \cdot \varepsilon_s + 0.15 \cdot f_{1.0}^{\mathbf{x}_0} + 0.25 \cdot f_{0.8}^{\mathbf{x}_0} \right) + \frac{1}{3} \cdot f_{0.6}^{\mathbf{x}_0} \tag{75}$$

$$= 0.4 \cdot \varepsilon_s + \frac{3}{30} \cdot f_{1.0}^{\mathbf{x}_0} + \frac{5}{30} \cdot f_{0.8}^{\mathbf{x}_0} + \frac{10}{30} \cdot f_{0.6} \tag{76}$$

$$= 0.4 \cdot \varepsilon_s + 0.6 \cdot \left( \frac{3}{18} \cdot f_{1.0}^{\mathbf{x}_0} + \frac{5}{18} \cdot f_{0.8}^{\mathbf{x}_0} + \frac{10}{18} \cdot f_{0.6} \right) \tag{77}$$

We now proceed to **the fourth** update:

$$\hat{\mathbf{x}}_{0.2} = \hat{\mathbf{x}}_{0.4} + (0.2 - 0.4) \cdot \mu_\theta(\hat{\mathbf{x}}_{0.4}, 0.4) \tag{78}$$

$$= \hat{\mathbf{x}}_{0.4} - 0.2 \cdot \frac{\hat{\mathbf{x}}_{0.4} - f_{0.4}^{\mathbf{x}_0}}{0.4} \tag{79}$$

$$= \frac{1}{2} \cdot \hat{\mathbf{x}}_{0.4} + \frac{1}{2} \cdot f_{0.4}^{\mathbf{x}_0} \tag{80}$$

$$= \frac{1}{2} \cdot \left( 0.4 \cdot \varepsilon_s + \frac{3}{30} \cdot f_{1.0}^{\mathbf{x}_0} + \frac{5}{30} \cdot f_{0.8}^{\mathbf{x}_0} + \frac{10}{30} \cdot f_{0.6} \right) + \frac{1}{2} \cdot f_{0.4}^{\mathbf{x}_0} \tag{81}$$

$$= 0.2 \cdot \varepsilon_s + \frac{3}{60} \cdot f_{1.0}^{\mathbf{x}_0} + \frac{5}{60} \cdot f_{0.8}^{\mathbf{x}_0} + \frac{10}{60} \cdot f_{0.6} + \frac{30}{60} \cdot f_{0.4}^{\mathbf{x}_0} \tag{82}$$

$$= 0.2 \cdot \varepsilon_s + 0.8 \cdot \left( \frac{3}{48} \cdot f_{1.0}^{\mathbf{x}_0} + \frac{5}{48} \cdot f_{0.8}^{\mathbf{x}_0} + \frac{10}{48} \cdot f_{0.6} + \frac{30}{48} \cdot f_{0.4}^{\mathbf{x}_0} \right) \tag{83}$$

We now proceed to **the fifth** update:

$$\hat{\mathbf{x}}_{0.0} = \hat{\mathbf{x}}_{0.2} + (0.4 - 0.2) \cdot \mu_\theta(\hat{\mathbf{x}}_{0.2}, 0.2) \tag{84}$$

$$= \hat{\mathbf{x}}_{0.2} - 0.2 \cdot \frac{\hat{\mathbf{x}}_{0.2} - f_{0.2}^{\mathbf{x}_0}}{0.2} \tag{85}$$

$$= 0.0 \cdot \varepsilon_s + 1.0 \cdot f_{0.2}^{\mathbf{x}_0} \tag{86}$$

It can be seen that the compositions of $\hat{\mathbf{x}}_{0.4}$, $\hat{\mathbf{x}}_{0.2}$, and $\hat{\mathbf{x}}_{0.0}$ all share the same structural properties.

Up to this point, we have expressed the Euler inference method in the form of Natural Inference.

## D   COEFFICIENT MATRIXES

### D.1   DDPM COEFFICIENT MATRIX

Table 3: DDPM's signal coefficient matrix on Natural Inference framework

| time | 999 | 940 | 881 | 823 | 764 | 705 | 646 | 588 | 529 | 470 | 411 | 353 | 294 | 235 | 176 | 118 | 059 | 000 | sum |
|---|---|---|---|---|---|---|---|---|---|---|---|---|---|---|---|---|---|---|---|
| 940 | 0.008 | 0.0 | 0.0 | 0.0 | 0.0 | 0.0 | 0.0 | 0.0 | 0.0 | 0.0 | 0.0 | 0.0 | 0.0 | 0.0 | 0.0 | 0.0 | 0.0 | 0.0 | 0.008 |
| 881 | 0.005 | 0.013 | 0.0 | 0.0 | 0.0 | 0.0 | 0.0 | 0.0 | 0.0 | 0.0 | 0.0 | 0.0 | 0.0 | 0.0 | 0.0 | 0.0 | 0.0 | 0.0 | 0.017 |
| 823 | 0.003 | 0.008 | 0.02 | 0.0 | 0.0 | 0.0 | 0.0 | 0.0 | 0.0 | 0.0 | 0.0 | 0.0 | 0.0 | 0.0 | 0.0 | 0.0 | 0.0 | 0.0 | 0.031 |
| 764 | 0.002 | 0.005 | 0.013 | 0.032 | 0.0 | 0.0 | 0.0 | 0.0 | 0.0 | 0.0 | 0.0 | 0.0 | 0.0 | 0.0 | 0.0 | 0.0 | 0.0 | 0.0 | 0.051 |
| 705 | 0.001 | 0.003 | 0.008 | 0.02 | 0.047 | 0.0 | 0.0 | 0.0 | 0.0 | 0.0 | 0.0 | 0.0 | 0.0 | 0.0 | 0.0 | 0.0 | 0.0 | 0.0 | 0.079 |
| 646 | 0.001 | 0.002 | 0.005 | 0.013 | 0.031 | 0.067 | 0.0 | 0.0 | 0.0 | 0.0 | 0.0 | 0.0 | 0.0 | 0.0 | 0.0 | 0.0 | 0.0 | 0.0 | 0.119 |
| 588 | 0.0 | 0.001 | 0.004 | 0.009 | 0.021 | 0.046 | 0.09 | 0.0 | 0.0 | 0.0 | 0.0 | 0.0 | 0.0 | 0.0 | 0.0 | 0.0 | 0.0 | 0.0 | 0.172 |
| 529 | 0.0 | 0.001 | 0.003 | 0.006 | 0.015 | 0.032 | 0.062 | 0.12 | 0.0 | 0.0 | 0.0 | 0.0 | 0.0 | 0.0 | 0.0 | 0.0 | 0.0 | 0.0 | 0.24 |
| 470 | 0.0 | 0.001 | 0.002 | 0.005 | 0.01 | 0.022 | 0.044 | 0.085 | 0.154 | 0.0 | 0.0 | 0.0 | 0.0 | 0.0 | 0.0 | 0.0 | 0.0 | 0.0 | 0.323 |
| 411 | 0.0 | 0.0 | 0.001 | 0.003 | 0.007 | 0.016 | 0.031 | 0.06 | 0.109 | 0.192 | 0.0 | 0.0 | 0.0 | 0.0 | 0.0 | 0.0 | 0.0 | 0.0 | 0.42 |
| 353 | 0.0 | 0.0 | 0.001 | 0.002 | 0.005 | 0.011 | 0.022 | 0.042 | 0.076 | 0.135 | 0.232 | 0.0 | 0.0 | 0.0 | 0.0 | 0.0 | 0.0 | 0.0 | 0.526 |
| 294 | 0.0 | 0.0 | 0.001 | 0.002 | 0.003 | 0.007 | 0.015 | 0.028 | 0.051 | 0.091 | 0.156 | 0.284 | 0.0 | 0.0 | 0.0 | 0.0 | 0.0 | 0.0 | 0.639 |
| 235 | 0.0 | 0.0 | 0.0 | 0.001 | 0.002 | 0.005 | 0.009 | 0.018 | 0.033 | 0.057 | 0.099 | 0.18 | 0.345 | 0.0 | 0.0 | 0.0 | 0.0 | 0.0 | 0.749 |
| 176 | 0.0 | 0.0 | 0.0 | 0.001 | 0.001 | 0.003 | 0.005 | 0.01 | 0.018 | 0.032 | 0.056 | 0.101 | 0.195 | 0.426 | 0.0 | 0.0 | 0.0 | 0.0 | 0.849 |
| 118 | 0.0 | 0.0 | 0.0 | 0.0 | 0.001 | 0.001 | 0.002 | 0.005 | 0.008 | 0.015 | 0.026 | 0.047 | 0.09 | 0.196 | 0.536 | 0.0 | 0.0 | 0.0 | 0.927 |
| 059 | 0.0 | 0.0 | 0.0 | 0.0 | 0.0 | 0.0 | 0.001 | 0.001 | 0.002 | 0.004 | 0.007 | 0.013 | 0.024 | 0.053 | 0.145 | 0.728 | 0.0 | 0.0 | 0.98 |
| 000 | 0.0 | 0.0 | 0.0 | 0.0 | 0.0 | 0.0 | 0.0 | 0.0 | 0.0 | 0.0 | 0.0 | 0.0 | 0.0 | 0.0 | 0.0 | 0.002 | 0.998 | 0.0 | 1.0 |
| -01 | 0.0 | 0.0 | 0.0 | 0.0 | 0.0 | 0.0 | 0.0 | 0.0 | 0.0 | 0.0 | 0.0 | 0.0 | 0.0 | 0.0 | 0.0 | 0.0 | 0.0 | 1.0 | 1.0 |

Table 4: DDPM's noise coefficient matrix on Natural Inference framework

| time | 999 | 940 | 881 | 823 | 764 | 705 | 646 | 588 | 529 | 470 | 411 | 353 | 294 | 235 | 176 | 118 | 059 | 000 | -01 | norm |
|---|---|---|---|---|---|---|---|---|---|---|---|---|---|---|---|---|---|---|---|---|
| 940 | 0.561 | 0.828 | 0.0 | 0.0 | 0.0 | 0.0 | 0.0 | 0.0 | 0.0 | 0.0 | 0.0 | 0.0 | 0.0 | 0.0 | 0.0 | 0.0 | 0.0 | 0.0 | 0.0 | 1.0 |
| 881 | 0.326 | 0.481 | 0.814 | 0.0 | 0.0 | 0.0 | 0.0 | 0.0 | 0.0 | 0.0 | 0.0 | 0.0 | 0.0 | 0.0 | 0.0 | 0.0 | 0.0 | 0.0 | 0.0 | 1.0 |
| 823 | 0.197 | 0.292 | 0.494 | 0.795 | 0.0 | 0.0 | 0.0 | 0.0 | 0.0 | 0.0 | 0.0 | 0.0 | 0.0 | 0.0 | 0.0 | 0.0 | 0.0 | 0.0 | 0.0 | 0.999 |
| 764 | 0.123 | 0.181 | 0.307 | 0.494 | 0.782 | 0.0 | 0.0 | 0.0 | 0.0 | 0.0 | 0.0 | 0.0 | 0.0 | 0.0 | 0.0 | 0.0 | 0.0 | 0.0 | 0.0 | 0.999 |
| 705 | 0.079 | 0.117 | 0.197 | 0.318 | 0.502 | 0.763 | 0.0 | 0.0 | 0.0 | 0.0 | 0.0 | 0.0 | 0.0 | 0.0 | 0.0 | 0.0 | 0.0 | 0.0 | 0.0 | 0.997 |
| 646 | 0.052 | 0.077 | 0.131 | 0.211 | 0.333 | 0.506 | 0.741 | 0.0 | 0.0 | 0.0 | 0.0 | 0.0 | 0.0 | 0.0 | 0.0 | 0.0 | 0.0 | 0.0 | 0.0 | 0.993 |
| 588 | 0.036 | 0.053 | 0.09 | 0.144 | 0.228 | 0.347 | 0.508 | 0.712 | 0.0 | 0.0 | 0.0 | 0.0 | 0.0 | 0.0 | 0.0 | 0.0 | 0.0 | 0.0 | 0.0 | 0.985 |
| 529 | 0.025 | 0.037 | 0.062 | 0.1 | 0.159 | 0.241 | 0.353 | 0.496 | 0.687 | 0.0 | 0.0 | 0.0 | 0.0 | 0.0 | 0.0 | 0.0 | 0.0 | 0.0 | 0.0 | 0.971 |
| 470 | 0.018 | 0.026 | 0.044 | 0.071 | 0.112 | 0.17 | 0.249 | 0.349 | 0.485 | 0.653 | 0.0 | 0.0 | 0.0 | 0.0 | 0.0 | 0.0 | 0.0 | 0.0 | 0.0 | 0.946 |
| 411 | 0.012 | 0.018 | 0.031 | 0.05 | 0.079 | 0.12 | 0.176 | 0.247 | 0.342 | 0.462 | 0.613 | 0.0 | 0.0 | 0.0 | 0.0 | 0.0 | 0.0 | 0.0 | 0.0 | 0.907 |
| 353 | 0.009 | 0.013 | 0.022 | 0.035 | 0.056 | 0.084 | 0.123 | 0.173 | 0.24 | 0.324 | 0.43 | 0.564 | 0.0 | 0.0 | 0.0 | 0.0 | 0.0 | 0.0 | 0.0 | 0.85 |
| 294 | 0.006 | 0.009 | 0.015 | 0.024 | 0.037 | 0.057 | 0.083 | 0.117 | 0.162 | 0.218 | 0.29 | 0.38 | 0.513 | 0.0 | 0.0 | 0.0 | 0.0 | 0.0 | 0.0 | 0.769 |
| 235 | 0.004 | 0.005 | 0.009 | 0.015 | 0.024 | 0.036 | 0.053 | 0.074 | 0.102 | 0.138 | 0.183 | 0.24 | 0.324 | 0.449 | 0.0 | 0.0 | 0.0 | 0.0 | 0.0 | 0.662 |
| 176 | 0.002 | 0.003 | 0.005 | 0.008 | 0.013 | 0.02 | 0.03 | 0.042 | 0.058 | 0.078 | 0.103 | 0.135 | 0.183 | 0.253 | 0.375 | 0.0 | 0.0 | 0.0 | 0.0 | 0.529 |
| 118 | 0.001 | 0.001 | 0.002 | 0.004 | 0.006 | 0.009 | 0.014 | 0.019 | 0.027 | 0.036 | 0.048 | 0.062 | 0.084 | 0.117 | 0.173 | 0.285 | 0.0 | 0.0 | 0.0 | 0.375 |
| 059 | 0.0 | 0.0 | 0.001 | 0.001 | 0.002 | 0.003 | 0.004 | 0.005 | 0.007 | 0.01 | 0.013 | 0.017 | 0.023 | 0.032 | 0.047 | 0.077 | 0.173 | 0.0 | 0.0 | 0.201 |
| 000 | 0.0 | 0.0 | 0.0 | 0.0 | 0.0 | 0.0 | 0.0 | 0.0 | 0.0 | 0.0 | 0.0 | 0.0 | 0.0 | 0.0 | 0.0 | 0.0 | 0.0 | 0.01 | 0.0 | 0.01 |
| -01 | 0.0 | 0.0 | 0.0 | 0.0 | 0.0 | 0.0 | 0.0 | 0.0 | 0.0 | 0.0 | 0.0 | 0.0 | 0.0 | 0.0 | 0.0 | 0.0 | 0.0 | 0.0 | 0.0 | 0.00 |

## D.2 DDIM COEFFICIENT MATRIX

Table 5: DDIM's signal coefficient matrix on the Natural Inference framework

| time | 999 | 940 | 881 | 823 | 764 | 705 | 646 | 588 | 529 | 470 | 411 | 353 | 294 | 235 | 176 | 118 | 059 | 000 | sum |
|---|---|---|---|---|---|---|---|---|---|---|---|---|---|---|---|---|---|---|---|
| 940 | 0.005 | 0.0 | 0.0 | 0.0 | 0.0 | 0.0 | 0.0 | 0.0 | 0.0 | 0.0 | 0.0 | 0.0 | 0.0 | 0.0 | 0.0 | 0.0 | 0.0 | 0.0 | 0.005 |
| 881 | 0.005 | 0.008 | 0.0 | 0.0 | 0.0 | 0.0 | 0.0 | 0.0 | 0.0 | 0.0 | 0.0 | 0.0 | 0.0 | 0.0 | 0.0 | 0.0 | 0.0 | 0.0 | 0.013 |
| 823 | 0.005 | 0.008 | 0.013 | 0.0 | 0.0 | 0.0 | 0.0 | 0.0 | 0.0 | 0.0 | 0.0 | 0.0 | 0.0 | 0.0 | 0.0 | 0.0 | 0.0 | 0.0 | 0.026 |
| 764 | 0.005 | 0.008 | 0.013 | 0.019 | 0.0 | 0.0 | 0.0 | 0.0 | 0.0 | 0.0 | 0.0 | 0.0 | 0.0 | 0.0 | 0.0 | 0.0 | 0.0 | 0.0 | 0.045 |
| 705 | 0.005 | 0.008 | 0.013 | 0.019 | 0.028 | 0.0 | 0.0 | 0.0 | 0.0 | 0.0 | 0.0 | 0.0 | 0.0 | 0.0 | 0.0 | 0.0 | 0.0 | 0.0 | 0.074 |
| 646 | 0.005 | 0.008 | 0.013 | 0.019 | 0.028 | 0.04 | 0.0 | 0.0 | 0.0 | 0.0 | 0.0 | 0.0 | 0.0 | 0.0 | 0.0 | 0.0 | 0.0 | 0.0 | 0.113 |
| 588 | 0.005 | 0.008 | 0.012 | 0.019 | 0.028 | 0.04 | 0.053 | 0.0 | 0.0 | 0.0 | 0.0 | 0.0 | 0.0 | 0.0 | 0.0 | 0.0 | 0.0 | 0.0 | 0.166 |
| 529 | 0.005 | 0.008 | 0.012 | 0.019 | 0.028 | 0.039 | 0.052 | 0.07 | 0.0 | 0.0 | 0.0 | 0.0 | 0.0 | 0.0 | 0.0 | 0.0 | 0.0 | 0.0 | 0.234 |
| 470 | 0.005 | 0.008 | 0.012 | 0.018 | 0.027 | 0.038 | 0.051 | 0.069 | 0.089 | 0.0 | 0.0 | 0.0 | 0.0 | 0.0 | 0.0 | 0.0 | 0.0 | 0.0 | 0.317 |
| 411 | 0.005 | 0.007 | 0.011 | 0.018 | 0.026 | 0.037 | 0.049 | 0.066 | 0.086 | 0.111 | 0.0 | 0.0 | 0.0 | 0.0 | 0.0 | 0.0 | 0.0 | 0.0 | 0.415 |
| 353 | 0.004 | 0.007 | 0.011 | 0.017 | 0.024 | 0.034 | 0.046 | 0.062 | 0.08 | 0.104 | 0.132 | 0.0 | 0.0 | 0.0 | 0.0 | 0.0 | 0.0 | 0.0 | 0.521 |
| 294 | 0.004 | 0.006 | 0.01 | 0.015 | 0.022 | 0.031 | 0.041 | 0.056 | 0.073 | 0.094 | 0.12 | 0.163 | 0.0 | 0.0 | 0.0 | 0.0 | 0.0 | 0.0 | 0.634 |
| 235 | 0.003 | 0.005 | 0.008 | 0.013 | 0.019 | 0.027 | 0.036 | 0.048 | 0.063 | 0.081 | 0.103 | 0.14 | 0.199 | 0.0 | 0.0 | 0.0 | 0.0 | 0.0 | 0.745 |
| 176 | 0.003 | 0.004 | 0.007 | 0.01 | 0.015 | 0.021 | 0.029 | 0.038 | 0.05 | 0.065 | 0.082 | 0.112 | 0.159 | 0.25 | 0.0 | 0.0 | 0.0 | 0.0 | 0.845 |
| 118 | 0.002 | 0.003 | 0.005 | 0.007 | 0.011 | 0.015 | 0.02 | 0.027 | 0.035 | 0.046 | 0.058 | 0.08 | 0.113 | 0.177 | 0.325 | 0.0 | 0.0 | 0.0 | 0.924 |
| 059 | 0.001 | 0.002 | 0.003 | 0.004 | 0.006 | 0.008 | 0.011 | 0.015 | 0.019 | 0.025 | 0.031 | 0.043 | 0.06 | 0.095 | 0.174 | 0.483 | 0.0 | 0.0 | 0.978 |
| 000 | 0.0 | 0.0 | 0.0 | 0.0 | 0.0 | 0.0 | 0.001 | 0.001 | 0.001 | 0.001 | 0.002 | 0.002 | 0.003 | 0.005 | 0.009 | 0.024 | 0.951 | 0.0 | 1.0 |
| -01 | 0.0 | 0.0 | 0.0 | 0.0 | 0.0 | 0.0 | 0.0 | 0.0 | 0.0 | 0.0 | 0.0 | 0.0 | 0.0 | 0.0 | 0.0 | 0.0 | 0.0 | 1.0 | 1.0 |

## D.3 FLOW MATCHING COEFFICIENT MATRIX

Table 6: Flow Matching Euler sampler's signal coefficient matrix on Natural Inference framework

| time | 1.000 | 0.944 | 0.889 | 0.833 | 0.778 | 0.722 | 0.667 | 0.611 | 0.556 | 0.500 | 0.444 | 0.389 | 0.333 | 0.278 | 0.222 | 0.167 | 0.111 | 0.056 | sum |
|------|-------|-------|-------|-------|-------|-------|-------|-------|-------|-------|-------|-------|-------|-------|-------|-------|-------|-------|-----|
| 0.944 | 0.056 | 0.0 | 0.0 | 0.0 | 0.0 | 0.0 | 0.0 | 0.0 | 0.0 | 0.0 | 0.0 | 0.0 | 0.0 | 0.0 | 0.0 | 0.0 | 0.0 | 0.0 | 0.056 |
| 0.889 | 0.052 | 0.059 | 0.0 | 0.0 | 0.0 | 0.0 | 0.0 | 0.0 | 0.0 | 0.0 | 0.0 | 0.0 | 0.0 | 0.0 | 0.0 | 0.0 | 0.0 | 0.0 | 0.111 |
| 0.833 | 0.049 | 0.055 | 0.062 | 0.0 | 0.0 | 0.0 | 0.0 | 0.0 | 0.0 | 0.0 | 0.0 | 0.0 | 0.0 | 0.0 | 0.0 | 0.0 | 0.0 | 0.0 | 0.167 |
| 0.778 | 0.046 | 0.051 | 0.058 | 0.067 | 0.0 | 0.0 | 0.0 | 0.0 | 0.0 | 0.0 | 0.0 | 0.0 | 0.0 | 0.0 | 0.0 | 0.0 | 0.0 | 0.0 | 0.222 |
| 0.722 | 0.042 | 0.048 | 0.054 | 0.062 | 0.071 | 0.0 | 0.0 | 0.0 | 0.0 | 0.0 | 0.0 | 0.0 | 0.0 | 0.0 | 0.0 | 0.0 | 0.0 | 0.0 | 0.278 |
| 0.667 | 0.039 | 0.044 | 0.05 | 0.057 | 0.066 | 0.077 | 0.0 | 0.0 | 0.0 | 0.0 | 0.0 | 0.0 | 0.0 | 0.0 | 0.0 | 0.0 | 0.0 | 0.0 | 0.333 |
| 0.611 | 0.036 | 0.04 | 0.046 | 0.052 | 0.06 | 0.071 | 0.083 | 0.0 | 0.0 | 0.0 | 0.0 | 0.0 | 0.0 | 0.0 | 0.0 | 0.0 | 0.0 | 0.0 | 0.389 |
| 0.556 | 0.033 | 0.037 | 0.042 | 0.048 | 0.055 | 0.064 | 0.076 | 0.091 | 0.0 | 0.0 | 0.0 | 0.0 | 0.0 | 0.0 | 0.0 | 0.0 | 0.0 | 0.0 | 0.444 |
| 0.500 | 0.029 | 0.033 | 0.038 | 0.043 | 0.049 | 0.058 | 0.068 | 0.082 | 0.1 | 0.0 | 0.0 | 0.0 | 0.0 | 0.0 | 0.0 | 0.0 | 0.0 | 0.0 | 0.5 |
| 0.444 | 0.026 | 0.029 | 0.033 | 0.038 | 0.044 | 0.051 | 0.061 | 0.073 | 0.089 | 0.111 | 0.0 | 0.0 | 0.0 | 0.0 | 0.0 | 0.0 | 0.0 | 0.0 | 0.556 |
| 0.389 | 0.023 | 0.026 | 0.029 | 0.033 | 0.038 | 0.045 | 0.053 | 0.064 | 0.078 | 0.097 | 0.125 | 0.0 | 0.0 | 0.0 | 0.0 | 0.0 | 0.0 | 0.0 | 0.611 |
| 0.333 | 0.02 | 0.022 | 0.025 | 0.029 | 0.033 | 0.038 | 0.045 | 0.055 | 0.067 | 0.083 | 0.107 | 0.143 | 0.0 | 0.0 | 0.0 | 0.0 | 0.0 | 0.0 | 0.667 |
| 0.278 | 0.016 | 0.018 | 0.021 | 0.024 | 0.027 | 0.032 | 0.038 | 0.045 | 0.056 | 0.069 | 0.089 | 0.119 | 0.167 | 0.0 | 0.0 | 0.0 | 0.0 | 0.0 | 0.722 |
| 0.222 | 0.013 | 0.015 | 0.017 | 0.019 | 0.022 | 0.026 | 0.03 | 0.036 | 0.044 | 0.056 | 0.071 | 0.095 | 0.133 | 0.2 | 0.0 | 0.0 | 0.0 | 0.0 | 0.778 |
| 0.167 | 0.01 | 0.011 | 0.012 | 0.014 | 0.016 | 0.019 | 0.023 | 0.027 | 0.033 | 0.042 | 0.054 | 0.071 | 0.1 | 0.15 | 0.25 | 0.0 | 0.0 | 0.0 | 0.833 |
| 0.111 | 0.007 | 0.007 | 0.008 | 0.01 | 0.011 | 0.013 | 0.015 | 0.018 | 0.022 | 0.028 | 0.036 | 0.048 | 0.067 | 0.1 | 0.167 | 0.333 | 0.0 | 0.0 | 0.889 |
| 0.056 | 0.003 | 0.004 | 0.004 | 0.005 | 0.005 | 0.006 | 0.008 | 0.009 | 0.011 | 0.014 | 0.018 | 0.024 | 0.033 | 0.05 | 0.083 | 0.167 | 0.5 | 0.0 | 0.944 |
| 0.000 | 0.0 | 0.0 | 0.0 | 0.0 | 0.0 | 0.0 | 0.0 | 0.0 | 0.0 | 0.0 | 0.0 | 0.0 | 0.0 | 0.0 | 0.0 | 0.0 | 0.0 | 1.0 | 1.0 |

## D.4 DEIS COEFFICIENT MATRIX

Table 7: DEIS sampler's signal coefficient matrix on Natural Inference framework

| time | 1.000 | 0.895 | 0.796 | 0.703 | 0.616 | 0.534 | 0.459 | 0.389 | 0.324 | 0.266 | 0.213 | 0.167 | 0.126 | 0.090 | 0.061 | 0.037 | 0.019 | 0.007 | sum |
|------|-------|-------|-------|-------|-------|-------|-------|-------|-------|-------|-------|-------|-------|-------|-------|-------|-------|-------|-----|
| 0.895 | 0.011 | 0.0 | 0.0 | 0.0 | 0.0 | 0.0 | 0.0 | 0.0 | 0.0 | 0.0 | 0.0 | 0.0 | 0.0 | 0.0 | 0.0 | 0.0 | 0.0 | 0.0 | 0.011 |
| 0.796 | 0.002 | 0.033 | 0.0 | 0.0 | 0.0 | 0.0 | 0.0 | 0.0 | 0.0 | 0.0 | 0.0 | 0.0 | 0.0 | 0.0 | 0.0 | 0.0 | 0.0 | 0.0 | 0.034 |
| 0.703 | 0.014 | -0.01 | 0.072 | 0.0 | 0.0 | 0.0 | 0.0 | 0.0 | 0.0 | 0.0 | 0.0 | 0.0 | 0.0 | 0.0 | 0.0 | 0.0 | 0.0 | 0.0 | 0.076 |
| 0.616 | -0.005 | 0.058 | -0.043 | 0.13 | 0.0 | 0.0 | 0.0 | 0.0 | 0.0 | 0.0 | 0.0 | 0.0 | 0.0 | 0.0 | 0.0 | 0.0 | 0.0 | 0.0 | 0.14 |
| 0.534 | 0.014 | -0.013 | 0.09 | -0.046 | 0.183 | 0.0 | 0.0 | 0.0 | 0.0 | 0.0 | 0.0 | 0.0 | 0.0 | 0.0 | 0.0 | 0.0 | 0.0 | 0.0 | 0.229 |
| 0.459 | -0.004 | 0.054 | -0.037 | 0.135 | -0.046 | 0.235 | 0.0 | 0.0 | 0.0 | 0.0 | 0.0 | 0.0 | 0.0 | 0.0 | 0.0 | 0.0 | 0.0 | 0.0 | 0.337 |
| 0.389 | 0.011 | -0.005 | 0.069 | -0.02 | 0.165 | -0.046 | 0.283 | 0.0 | 0.0 | 0.0 | 0.0 | 0.0 | 0.0 | 0.0 | 0.0 | 0.0 | 0.0 | 0.0 | 0.457 |
| 0.324 | -0.001 | 0.038 | -0.015 | 0.093 | -0.004 | 0.19 | -0.047 | 0.324 | 0.0 | 0.0 | 0.0 | 0.0 | 0.0 | 0.0 | 0.0 | 0.0 | 0.0 | 0.0 | 0.577 |
| 0.266 | 0.007 | 0.004 | 0.041 | 0.004 | 0.105 | 0.009 | 0.209 | -0.053 | 0.363 | 0.0 | 0.0 | 0.0 | 0.0 | 0.0 | 0.0 | 0.0 | 0.0 | 0.0 | 0.689 |
| 0.213 | 0.001 | 0.023 | -0.001 | 0.055 | 0.017 | 0.113 | 0.016 | 0.223 | -0.063 | 0.401 | 0.0 | 0.0 | 0.0 | 0.0 | 0.0 | 0.0 | 0.0 | 0.0 | 0.785 |
| 0.167 | 0.004 | 0.006 | 0.022 | 0.012 | 0.06 | 0.025 | 0.116 | 0.015 | 0.234 | -0.076 | 0.441 | 0.0 | 0.0 | 0.0 | 0.0 | 0.0 | 0.0 | 0.0 | 0.86 |
| 0.126 | 0.001 | 0.013 | 0.003 | 0.03 | 0.018 | 0.064 | 0.026 | 0.117 | 0.009 | 0.245 | -0.094 | 0.487 | 0.0 | 0.0 | 0.0 | 0.0 | 0.0 | 0.0 | 0.916 |
| 0.090 | 0.002 | 0.005 | 0.011 | 0.01 | 0.032 | 0.02 | 0.06 | 0.021 | 0.115 | -0.003 | 0.257 | -0.115 | 0.541 | 0.0 | 0.0 | 0.0 | 0.0 | 0.0 | 0.954 |
| 0.061 | 0.001 | 0.006 | 0.002 | 0.015 | 0.011 | 0.03 | 0.016 | 0.056 | 0.012 | 0.114 | -0.02 | 0.271 | -0.141 | 0.606 | 0.0 | 0.0 | 0.0 | 0.0 | 0.977 |
| 0.037 | 0.001 | 0.002 | 0.005 | 0.004 | 0.014 | 0.009 | 0.027 | 0.01 | 0.051 | -0.0 | 0.112 | -0.042 | 0.284 | -0.173 | 0.687 | 0.0 | 0.0 | 0.0 | 0.99 |
| 0.019 | 0.0 | 0.002 | 0.001 | 0.006 | 0.004 | 0.012 | 0.005 | 0.022 | 0.002 | 0.045 | -0.014 | 0.11 | -0.066 | 0.292 | -0.208 | 0.785 | 0.0 | 0.0 | 0.997 |
| 0.007 | 0.0 | 0.0 | 0.002 | 0.001 | 0.004 | 0.002 | 0.008 | 0.001 | 0.017 | -0.005 | 0.039 | -0.027 | 0.103 | -0.088 | 0.285 | -0.244 | 0.902 | 0.0 | 0.999 |
| 0.001 | -0.0 | 0.0 | -0.0 | 0.001 | -0.0 | 0.002 | -0.001 | 0.005 | -0.003 | 0.012 | -0.012 | 0.033 | -0.039 | 0.09 | -0.111 | 0.262 | -0.319 | 1.078 | 1.0 |

## D.5 DPMSOLVER COEFFICIENT MATRIX

Table 8: DPMSolver2S's signal coefficient matrix on Natural Inference framework

| time | 1.000 | 0.946 | 0.889 | 0.835 | 0.778 | 0.724 | 0.667 | 0.614 | 0.556 | 0.502 | 0.445 | 0.390 | 0.334 | 0.277 | 0.223 | 0.161 | 0.112 | 0.016 | sum |
|------|-------|-------|-------|-------|-------|-------|-------|-------|-------|-------|-------|-------|-------|-------|-------|-------|-------|-------|-----|
| 0.946 | 0.005 | 0.0 | 0.0 | 0.0 | 0.0 | 0.0 | 0.0 | 0.0 | 0.0 | 0.0 | 0.0 | 0.0 | 0.0 | 0.0 | 0.0 | 0.0 | 0.0 | 0.0 | 0.005 |
| 0.889 | -0.008 | 0.021 | 0.0 | 0.0 | 0.0 | 0.0 | 0.0 | 0.0 | 0.0 | 0.0 | 0.0 | 0.0 | 0.0 | 0.0 | 0.0 | 0.0 | 0.0 | 0.0 | 0.012 |
| 0.835 | -0.008 | 0.021 | 0.011 | 0.0 | 0.0 | 0.0 | 0.0 | 0.0 | 0.0 | 0.0 | 0.0 | 0.0 | 0.0 | 0.0 | 0.0 | 0.0 | 0.0 | 0.0 | 0.023 |
| 0.778 | -0.008 | 0.021 | -0.017 | 0.045 | 0.0 | 0.0 | 0.0 | 0.0 | 0.0 | 0.0 | 0.0 | 0.0 | 0.0 | 0.0 | 0.0 | 0.0 | 0.0 | 0.0 | 0.041 |
| 0.724 | -0.008 | 0.021 | -0.017 | 0.045 | 0.024 | 0.0 | 0.0 | 0.0 | 0.0 | 0.0 | 0.0 | 0.0 | 0.0 | 0.0 | 0.0 | 0.0 | 0.0 | 0.0 | 0.064 |
| 0.667 | -0.008 | 0.02 | -0.017 | 0.045 | -0.029 | 0.088 | 0.0 | 0.0 | 0.0 | 0.0 | 0.0 | 0.0 | 0.0 | 0.0 | 0.0 | 0.0 | 0.0 | 0.0 | 0.099 |
| 0.614 | -0.008 | 0.02 | -0.017 | 0.045 | -0.029 | 0.087 | 0.044 | 0.0 | 0.0 | 0.0 | 0.0 | 0.0 | 0.0 | 0.0 | 0.0 | 0.0 | 0.0 | 0.0 | 0.143 |
| 0.556 | -0.008 | 0.02 | -0.017 | 0.045 | -0.029 | 0.086 | -0.044 | 0.149 | 0.0 | 0.0 | 0.0 | 0.0 | 0.0 | 0.0 | 0.0 | 0.0 | 0.0 | 0.0 | 0.203 |
| 0.502 | -0.008 | 0.02 | -0.016 | 0.044 | -0.028 | 0.085 | -0.043 | 0.146 | 0.073 | 0.0 | 0.0 | 0.0 | 0.0 | 0.0 | 0.0 | 0.0 | 0.0 | 0.0 | 0.272 |
| 0.445 | -0.008 | 0.019 | -0.016 | 0.042 | -0.027 | 0.082 | -0.042 | 0.142 | -0.059 | 0.225 | 0.0 | 0.0 | 0.0 | 0.0 | 0.0 | 0.0 | 0.0 | 0.0 | 0.359 |
| 0.390 | -0.007 | 0.018 | -0.015 | 0.04 | -0.026 | 0.078 | -0.04 | 0.135 | -0.056 | 0.215 | 0.112 | 0.0 | 0.0 | 0.0 | 0.0 | 0.0 | 0.0 | 0.0 | 0.454 |
| 0.334 | -0.007 | 0.017 | -0.014 | 0.038 | -0.024 | 0.073 | -0.037 | 0.126 | -0.052 | 0.2 | -0.077 | 0.318 | 0.0 | 0.0 | 0.0 | 0.0 | 0.0 | 0.0 | 0.559 |
| 0.277 | -0.006 | 0.015 | -0.012 | 0.034 | -0.022 | 0.065 | -0.033 | 0.112 | -0.047 | 0.179 | -0.069 | 0.285 | 0.168 | 0.0 | 0.0 | 0.0 | 0.0 | 0.0 | 0.669 |
| 0.223 | -0.005 | 0.013 | -0.011 | 0.029 | -0.019 | 0.056 | -0.028 | 0.097 | -0.04 | 0.154 | -0.059 | 0.245 | -0.112 | 0.45 | 0.0 | 0.0 | 0.0 | 0.0 | 0.768 |
| 0.161 | -0.004 | 0.01 | -0.008 | 0.022 | -0.014 | 0.043 | -0.022 | 0.074 | -0.031 | 0.118 | -0.046 | 0.188 | -0.086 | 0.346 | 0.278 | 0.0 | 0.0 | 0.0 | 0.869 |
| 0.112 | -0.003 | 0.007 | -0.006 | 0.016 | -0.01 | 0.031 | -0.016 | 0.054 | -0.023 | 0.086 | -0.033 | 0.137 | -0.063 | 0.252 | -0.235 | 0.735 | 0.0 | 0.0 | 0.932 |
| 0.016 | -0.001 | 0.001 | -0.001 | 0.003 | -0.002 | 0.006 | -0.003 | 0.01 | -0.004 | 0.015 | -0.006 | 0.024 | -0.011 | 0.045 | -0.042 | 0.13 | 0.833 | 0.0 | 0.998 |
| 0.001 | -0.0 | 0.0 | -0.0 | 0.0 | -0.0 | 0.001 | -0.0 | 0.002 | -0.001 | 0.003 | -0.001 | 0.004 | -0.002 | 0.007 | -0.007 | 0.022 | -4.895 | 5.867 | 1.0 |

Table 9: DPMSolver3S's signal coefficient matrix on Natural Inference framework

| time | 1.000 | 0.948 | 0.892 | 0.834 | 0.782 | 0.727 | 0.667 | 0.615 | 0.560 | 0.500 | 0.447 | 0.391 | 0.334 | 0.273 | 0.217 | 0.167 | 0.044 | 0.009 | sum |
|---|---|---|---|---|---|---|---|---|---|---|---|---|---|---|---|---|---|---|---|
| 0.948 | 0.004 | 0.0 | 0.0 | 0.0 | 0.0 | 0.0 | 0.0 | 0.0 | 0.0 | 0.0 | 0.0 | 0.0 | 0.0 | 0.0 | 0.0 | 0.0 | 0.0 | 0.0 | 0.004 |
| 0.892 | -0.004 | 0.016 | 0.0 | 0.0 | 0.0 | 0.0 | 0.0 | 0.0 | 0.0 | 0.0 | 0.0 | 0.0 | 0.0 | 0.0 | 0.0 | 0.0 | 0.0 | 0.0 | 0.012 |
| 0.834 | 0.019 | -0.033 | 0.037 | 0.0 | 0.0 | 0.0 | 0.0 | 0.0 | 0.0 | 0.0 | 0.0 | 0.0 | 0.0 | 0.0 | 0.0 | 0.0 | 0.0 | 0.0 | 0.024 |
| 0.782 | 0.019 | -0.033 | 0.037 | 0.016 | 0.0 | 0.0 | 0.0 | 0.0 | 0.0 | 0.0 | 0.0 | 0.0 | 0.0 | 0.0 | 0.0 | 0.0 | 0.0 | 0.0 | 0.039 |
| 0.727 | 0.019 | -0.033 | 0.037 | -0.012 | 0.052 | 0.0 | 0.0 | 0.0 | 0.0 | 0.0 | 0.0 | 0.0 | 0.0 | 0.0 | 0.0 | 0.0 | 0.0 | 0.0 | 0.063 |
| 0.667 | 0.019 | -0.033 | 0.037 | 0.049 | -0.078 | 0.104 | 0.0 | 0.0 | 0.0 | 0.0 | 0.0 | 0.0 | 0.0 | 0.0 | 0.0 | 0.0 | 0.0 | 0.0 | 0.099 |
| 0.615 | 0.019 | -0.033 | 0.037 | 0.049 | -0.077 | 0.104 | 0.042 | 0.0 | 0.0 | 0.0 | 0.0 | 0.0 | 0.0 | 0.0 | 0.0 | 0.0 | 0.0 | 0.0 | 0.141 |
| 0.560 | 0.019 | -0.032 | 0.036 | 0.048 | -0.076 | 0.103 | -0.024 | 0.125 | 0.0 | 0.0 | 0.0 | 0.0 | 0.0 | 0.0 | 0.0 | 0.0 | 0.0 | 0.0 | 0.198 |
| 0.500 | 0.019 | -0.032 | 0.036 | 0.047 | -0.075 | 0.101 | 0.093 | -0.134 | 0.219 | 0.0 | 0.0 | 0.0 | 0.0 | 0.0 | 0.0 | 0.0 | 0.0 | 0.0 | 0.274 |
| 0.447 | 0.018 | -0.031 | 0.035 | 0.046 | -0.073 | 0.098 | 0.09 | -0.13 | 0.213 | 0.089 | 0.0 | 0.0 | 0.0 | 0.0 | 0.0 | 0.0 | 0.0 | 0.0 | 0.356 |
| 0.391 | 0.017 | -0.029 | 0.033 | 0.044 | -0.069 | 0.093 | 0.086 | -0.124 | 0.203 | -0.04 | 0.238 | 0.0 | 0.0 | 0.0 | 0.0 | 0.0 | 0.0 | 0.0 | 0.452 |
| 0.334 | 0.016 | -0.027 | 0.031 | 0.041 | -0.064 | 0.087 | 0.08 | -0.115 | 0.188 | 0.147 | -0.191 | 0.368 | 0.0 | 0.0 | 0.0 | 0.0 | 0.0 | 0.0 | 0.559 |
| 0.273 | 0.014 | -0.024 | 0.027 | 0.036 | -0.057 | 0.077 | 0.071 | -0.102 | 0.167 | 0.131 | -0.17 | 0.327 | 0.178 | 0.0 | 0.0 | 0.0 | 0.0 | 0.0 | 0.675 |
| 0.217 | 0.012 | -0.02 | 0.023 | 0.031 | -0.049 | 0.065 | 0.06 | -0.087 | 0.142 | 0.111 | -0.144 | 0.277 | -0.078 | 0.435 | 0.0 | 0.0 | 0.0 | 0.0 | 0.778 |
| 0.167 | 0.01 | -0.017 | 0.019 | 0.025 | -0.039 | 0.053 | 0.049 | -0.07 | 0.115 | 0.09 | -0.117 | 0.225 | 0.248 | -0.336 | 0.605 | 0.0 | 0.0 | 0.0 | 0.859 |
| 0.044 | 0.003 | -0.005 | 0.006 | 0.007 | -0.012 | 0.016 | 0.015 | -0.021 | 0.035 | 0.027 | -0.035 | 0.068 | 0.074 | -0.101 | 0.181 | 0.73 | 0.0 | 0.0 | 0.987 |
| 0.009 | 0.001 | -0.001 | 0.001 | 0.002 | -0.003 | 0.004 | 0.004 | -0.006 | 0.009 | 0.007 | -0.009 | 0.018 | 0.02 | -0.027 | 0.048 | -1.201 | 2.132 | 0.0 | 0.999 |
| 0.001 | 0.0 | -0.0 | 0.0 | 0.001 | -0.001 | 0.001 | 0.001 | -0.001 | 0.002 | 0.002 | -0.002 | 0.005 | 0.005 | -0.007 | 0.013 | 6.607 | -10.588 | 4.963 | 1.0 |

## D.6 DPMSolver++ coefficient matrix

Table 10: DPMSolverpp2S's signal coefficient matrix on Natural Inference framework

| time | 1.000 | 0.946 | 0.889 | 0.835 | 0.778 | 0.724 | 0.667 | 0.614 | 0.556 | 0.502 | 0.445 | 0.390 | 0.334 | 0.277 | 0.223 | 0.161 | 0.112 | 0.016 | sum |
|---|---|---|---|---|---|---|---|---|---|---|---|---|---|---|---|---|---|---|---|
| 0.946 | 0.005 | 0.0 | 0.0 | 0.0 | 0.0 | 0.0 | 0.0 | 0.0 | 0.0 | 0.0 | 0.0 | 0.0 | 0.0 | 0.0 | 0.0 | 0.0 | 0.0 | 0.0 | 0.005 |
| 0.889 | 0.0 | 0.012 | 0.0 | 0.0 | 0.0 | 0.0 | 0.0 | 0.0 | 0.0 | 0.0 | 0.0 | 0.0 | 0.0 | 0.0 | 0.0 | 0.0 | 0.0 | 0.0 | 0.012 |
| 0.835 | 0.0 | 0.012 | 0.011 | 0.0 | 0.0 | 0.0 | 0.0 | 0.0 | 0.0 | 0.0 | 0.0 | 0.0 | 0.0 | 0.0 | 0.0 | 0.0 | 0.0 | 0.0 | 0.023 |
| 0.778 | 0.0 | 0.012 | 0.0 | 0.029 | 0.0 | 0.0 | 0.0 | 0.0 | 0.0 | 0.0 | 0.0 | 0.0 | 0.0 | 0.0 | 0.0 | 0.0 | 0.0 | 0.0 | 0.041 |
| 0.724 | 0.0 | 0.012 | 0.0 | 0.029 | 0.024 | 0.0 | 0.0 | 0.0 | 0.0 | 0.0 | 0.0 | 0.0 | 0.0 | 0.0 | 0.0 | 0.0 | 0.0 | 0.0 | 0.064 |
| 0.667 | 0.0 | 0.012 | 0.0 | 0.028 | 0.0 | 0.059 | 0.0 | 0.0 | 0.0 | 0.0 | 0.0 | 0.0 | 0.0 | 0.0 | 0.0 | 0.0 | 0.0 | 0.0 | 0.099 |
| 0.614 | 0.0 | 0.012 | 0.0 | 0.028 | 0.0 | 0.058 | 0.044 | 0.0 | 0.0 | 0.0 | 0.0 | 0.0 | 0.0 | 0.0 | 0.0 | 0.0 | 0.0 | 0.0 | 0.143 |
| 0.556 | 0.0 | 0.012 | 0.0 | 0.028 | 0.0 | 0.058 | 0.0 | 0.105 | 0.0 | 0.0 | 0.0 | 0.0 | 0.0 | 0.0 | 0.0 | 0.0 | 0.0 | 0.0 | 0.203 |
| 0.502 | 0.0 | 0.012 | 0.0 | 0.028 | 0.0 | 0.057 | 0.0 | 0.103 | 0.073 | 0.0 | 0.0 | 0.0 | 0.0 | 0.0 | 0.0 | 0.0 | 0.0 | 0.0 | 0.272 |
| 0.445 | 0.0 | 0.011 | 0.0 | 0.027 | 0.0 | 0.055 | 0.0 | 0.1 | 0.0 | 0.166 | 0.0 | 0.0 | 0.0 | 0.0 | 0.0 | 0.0 | 0.0 | 0.0 | 0.359 |
| 0.390 | 0.0 | 0.011 | 0.0 | 0.025 | 0.0 | 0.052 | 0.0 | 0.095 | 0.0 | 0.159 | 0.112 | 0.0 | 0.0 | 0.0 | 0.0 | 0.0 | 0.0 | 0.0 | 0.454 |
| 0.334 | 0.0 | 0.01 | 0.0 | 0.024 | 0.0 | 0.049 | 0.0 | 0.089 | 0.0 | 0.147 | 0.0 | 0.241 | 0.0 | 0.0 | 0.0 | 0.0 | 0.0 | 0.0 | 0.559 |
| 0.277 | 0.0 | 0.009 | 0.0 | 0.021 | 0.0 | 0.044 | 0.0 | 0.079 | 0.0 | 0.132 | 0.0 | 0.216 | 0.168 | 0.0 | 0.0 | 0.0 | 0.0 | 0.0 | 0.669 |
| 0.223 | 0.0 | 0.008 | 0.0 | 0.018 | 0.0 | 0.037 | 0.0 | 0.068 | 0.0 | 0.113 | 0.0 | 0.185 | 0.0 | 0.338 | 0.0 | 0.0 | 0.0 | 0.0 | 0.768 |
| 0.161 | 0.0 | 0.006 | 0.0 | 0.014 | 0.0 | 0.029 | 0.0 | 0.052 | 0.0 | 0.087 | 0.0 | 0.143 | 0.0 | 0.26 | 0.278 | 0.0 | 0.0 | 0.0 | 0.869 |
| 0.112 | 0.0 | 0.004 | 0.0 | 0.01 | 0.0 | 0.021 | 0.0 | 0.038 | 0.0 | 0.064 | 0.0 | 0.104 | 0.0 | 0.189 | 0.0 | 0.501 | 0.0 | 0.0 | 0.932 |
| 0.016 | 0.0 | 0.001 | 0.0 | 0.002 | 0.0 | 0.004 | 0.0 | 0.007 | 0.0 | 0.011 | 0.0 | 0.018 | 0.0 | 0.034 | 0.0 | 0.089 | 0.833 | 0.0 | 0.998 |
| 0.001 | 0.0 | 0.0 | 0.0 | 0.0 | 0.0 | 0.001 | 0.0 | 0.001 | 0.0 | 0.002 | 0.0 | 0.003 | 0.0 | 0.006 | 0.0 | 0.015 | 0.0 | 0.972 | 1.0 |

Table 11: DPMSolverpp3S's signal coefficient matrix on Natural Inference framework

| time | 1.000 | 0.948 | 0.892 | 0.834 | 0.782 | 0.727 | 0.667 | 0.615 | 0.560 | 0.500 | 0.447 | 0.391 | 0.334 | 0.273 | 0.217 | 0.167 | 0.044 | 0.009 | sum |
|---|---|---|---|---|---|---|---|---|---|---|---|---|---|---|---|---|---|---|---|
| 0.948 | 0.004 | 0.0 | 0.0 | 0.0 | 0.0 | 0.0 | 0.0 | 0.0 | 0.0 | 0.0 | 0.0 | 0.0 | 0.0 | 0.0 | 0.0 | 0.0 | 0.0 | 0.0 | 0.004 |
| 0.892 | 0.025 | -0.014 | 0.0 | 0.0 | 0.0 | 0.0 | 0.0 | 0.0 | 0.0 | 0.0 | 0.0 | 0.0 | 0.0 | 0.0 | 0.0 | 0.0 | 0.0 | 0.0 | 0.012 |
| 0.834 | 0.046 | 0.0 | -0.022 | 0.0 | 0.0 | 0.0 | 0.0 | 0.0 | 0.0 | 0.0 | 0.0 | 0.0 | 0.0 | 0.0 | 0.0 | 0.0 | 0.0 | 0.0 | 0.024 |
| 0.782 | 0.046 | 0.0 | -0.022 | 0.016 | 0.0 | 0.0 | 0.0 | 0.0 | 0.0 | 0.0 | 0.0 | 0.0 | 0.0 | 0.0 | 0.0 | 0.0 | 0.0 | 0.0 | 0.039 |
| 0.727 | 0.046 | 0.0 | -0.022 | 0.085 | -0.045 | 0.0 | 0.0 | 0.0 | 0.0 | 0.0 | 0.0 | 0.0 | 0.0 | 0.0 | 0.0 | 0.0 | 0.0 | 0.0 | 0.063 |
| 0.667 | 0.046 | 0.0 | -0.022 | 0.144 | 0.0 | -0.068 | 0.0 | 0.0 | 0.0 | 0.0 | 0.0 | 0.0 | 0.0 | 0.0 | 0.0 | 0.0 | 0.0 | 0.0 | 0.099 |
| 0.615 | 0.045 | 0.0 | -0.022 | 0.143 | 0.0 | -0.068 | 0.042 | 0.0 | 0.0 | 0.0 | 0.0 | 0.0 | 0.0 | 0.0 | 0.0 | 0.0 | 0.0 | 0.0 | 0.141 |
| 0.560 | 0.045 | 0.0 | -0.022 | 0.142 | 0.0 | -0.067 | 0.211 | -0.111 | 0.0 | 0.0 | 0.0 | 0.0 | 0.0 | 0.0 | 0.0 | 0.0 | 0.0 | 0.0 | 0.198 |
| 0.500 | 0.044 | 0.0 | -0.021 | 0.139 | 0.0 | -0.066 | 0.334 | 0.0 | -0.156 | 0.0 | 0.0 | 0.0 | 0.0 | 0.0 | 0.0 | 0.0 | 0.0 | 0.0 | 0.274 |
| 0.447 | 0.043 | 0.0 | -0.021 | 0.135 | 0.0 | -0.064 | 0.325 | 0.0 | -0.151 | 0.089 | 0.0 | 0.0 | 0.0 | 0.0 | 0.0 | 0.0 | 0.0 | 0.0 | 0.356 |
| 0.391 | 0.041 | 0.0 | -0.02 | 0.129 | 0.0 | -0.061 | 0.31 | 0.0 | -0.144 | 0.415 | -0.217 | 0.0 | 0.0 | 0.0 | 0.0 | 0.0 | 0.0 | 0.0 | 0.452 |
| 0.334 | 0.038 | 0.0 | -0.018 | 0.119 | 0.0 | -0.057 | 0.288 | 0.0 | -0.134 | 0.6 | 0.0 | -0.277 | 0.0 | 0.0 | 0.0 | 0.0 | 0.0 | 0.0 | 0.559 |
| 0.273 | 0.034 | 0.0 | -0.016 | 0.106 | 0.0 | -0.05 | 0.255 | 0.0 | -0.119 | 0.533 | 0.0 | -0.246 | 0.178 | 0.0 | 0.0 | 0.0 | 0.0 | 0.0 | 0.675 |
| 0.217 | 0.029 | 0.0 | -0.014 | 0.09 | 0.0 | -0.043 | 0.217 | 0.0 | -0.101 | 0.452 | 0.0 | -0.209 | 0.749 | -0.393 | 0.0 | 0.0 | 0.0 | 0.0 | 0.778 |
| 0.167 | 0.023 | 0.0 | -0.011 | 0.073 | 0.0 | -0.035 | 0.176 | 0.0 | -0.082 | 0.368 | 0.0 | -0.17 | 0.962 | 0.0 | -0.445 | 0.0 | 0.0 | 0.0 | 0.859 |
| 0.044 | 0.007 | 0.0 | -0.003 | 0.022 | 0.0 | -0.01 | 0.053 | 0.0 | -0.025 | 0.11 | 0.0 | -0.051 | 0.288 | 0.0 | -0.133 | 0.73 | 0.0 | 0.0 | 0.987 |
| 0.009 | 0.002 | 0.0 | -0.001 | 0.006 | 0.0 | -0.003 | 0.014 | 0.0 | -0.007 | 0.029 | 0.0 | -0.013 | 0.076 | 0.0 | -0.035 | 2.235 | -1.304 | 0.0 | 0.999 |
| 0.001 | 0.0 | 0.0 | -0.0 | 0.002 | 0.0 | -0.001 | 0.004 | 0.0 | -0.002 | 0.008 | 0.0 | -0.004 | 0.02 | 0.0 | -0.009 | 2.116 | 0.0 | -1.134 | 1.0 |

# E  SD3's coefficient matrix and inference process visualization

## E.1  Coefficient matrix and its corresponding outputs

Note that, for readability, the coefficients in Table 12 are the original coefficients multiplied by 100. When using them, they should be normalized to the corresponding Marginal Coefficient for each step.

Table 12: SD3's signal coefficient matrix for Flow Matching Euler sampling

| time | 1.00 | 0.99 | 0.97 | 0.96 | 0.95 | 0.93 | 0.91 | 0.90 | 0.88 | 0.86 | 0.84 | 0.81 | 0.79 | 0.76 | 0.74 | 0.71 | 0.68 | 0.64 | 0.60 | 0.56 | 0.52 | 0.46 | 0.41 | 0.35 | 0.28 | 0.20 | 0.11 | 0.01 |
|---|---|---|---|---|---|---|---|---|---|---|---|---|---|---|---|---|---|---|---|---|---|---|---|---|---|---|---|---|
| 0.99 | 1.26 | 0.0 | 0.0 | 0.0 | 0.0 | 0.0 | 0.0 | 0.0 | 0.0 | 0.0 | 0.0 | 0.0 | 0.0 | 0.0 | 0.0 | 0.0 | 0.0 | 0.0 | 0.0 | 0.0 | 0.0 | 0.0 | 0.0 | 0.0 | 0.0 | 0.0 | 0.0 | 0.0 |
| 0.97 | 1.26 | 1.33 | 0.0 | 0.0 | 0.0 | 0.0 | 0.0 | 0.0 | 0.0 | 0.0 | 0.0 | 0.0 | 0.0 | 0.0 | 0.0 | 0.0 | 0.0 | 0.0 | 0.0 | 0.0 | 0.0 | 0.0 | 0.0 | 0.0 | 0.0 | 0.0 | 0.0 | 0.0 |
| 0.96 | 1.26 | 1.33 | 1.4 | 0.0 | 0.0 | 0.0 | 0.0 | 0.0 | 0.0 | 0.0 | 0.0 | 0.0 | 0.0 | 0.0 | 0.0 | 0.0 | 0.0 | 0.0 | 0.0 | 0.0 | 0.0 | 0.0 | 0.0 | 0.0 | 0.0 | 0.0 | 0.0 | 0.0 |
| 0.95 | 1.26 | 1.33 | 1.4 | 1.47 | 0.0 | 0.0 | 0.0 | 0.0 | 0.0 | 0.0 | 0.0 | 0.0 | 0.0 | 0.0 | 0.0 | 0.0 | 0.0 | 0.0 | 0.0 | 0.0 | 0.0 | 0.0 | 0.0 | 0.0 | 0.0 | 0.0 | 0.0 | 0.0 |
| 0.93 | 1.26 | 1.33 | 1.4 | 1.47 | 1.56 | 0.0 | 0.0 | 0.0 | 0.0 | 0.0 | 0.0 | 0.0 | 0.0 | 0.0 | 0.0 | 0.0 | 0.0 | 0.0 | 0.0 | 0.0 | 0.0 | 0.0 | 0.0 | 0.0 | 0.0 | 0.0 | 0.0 | 0.0 |
| 0.91 | 1.26 | 1.33 | 1.4 | 1.47 | 1.56 | 1.65 | 0.0 | 0.0 | 0.0 | 0.0 | 0.0 | 0.0 | 0.0 | 0.0 | 0.0 | 0.0 | 0.0 | 0.0 | 0.0 | 0.0 | 0.0 | 0.0 | 0.0 | 0.0 | 0.0 | 0.0 | 0.0 | 0.0 |
| 0.90 | 1.26 | 1.33 | 1.4 | 1.47 | 1.56 | 1.65 | 1.74 | 0.0 | 0.0 | 0.0 | 0.0 | 0.0 | 0.0 | 0.0 | 0.0 | 0.0 | 0.0 | 0.0 | 0.0 | 0.0 | 0.0 | 0.0 | 0.0 | 0.0 | 0.0 | 0.0 | 0.0 | 0.0 |
| 0.88 | 1.26 | 1.33 | 1.4 | 1.47 | 1.56 | 1.65 | 1.74 | 1.85 | 0.0 | 0.0 | 0.0 | 0.0 | 0.0 | 0.0 | 0.0 | 0.0 | 0.0 | 0.0 | 0.0 | 0.0 | 0.0 | 0.0 | 0.0 | 0.0 | 0.0 | 0.0 | 0.0 | 0.0 |
| 0.86 | 1.26 | 1.33 | 1.4 | 1.47 | 1.56 | 1.65 | 1.74 | 1.85 | 1.97 | 0.0 | 0.0 | 0.0 | 0.0 | 0.0 | 0.0 | 0.0 | 0.0 | 0.0 | 0.0 | 0.0 | 0.0 | 0.0 | 0.0 | 0.0 | 0.0 | 0.0 | 0.0 | 0.0 |
| 0.84 | 1.26 | 1.33 | 1.4 | 1.47 | 1.56 | 1.65 | 1.74 | 1.85 | 1.97 | 2.1 | 0.0 | 0.0 | 0.0 | 0.0 | 0.0 | 0.0 | 0.0 | 0.0 | 0.0 | 0.0 | 0.0 | 0.0 | 0.0 | 0.0 | 0.0 | 0.0 | 0.0 | 0.0 |
| 0.81 | 1.26 | 1.33 | 1.4 | 1.47 | 1.56 | 1.65 | 1.74 | 1.85 | 1.97 | 2.1 | 2.24 | 0.0 | 0.0 | 0.0 | 0.0 | 0.0 | 0.0 | 0.0 | 0.0 | 0.0 | 0.0 | 0.0 | 0.0 | 0.0 | 0.0 | 0.0 | 0.0 | 0.0 |
| 0.79 | 1.26 | 1.33 | 1.4 | 1.47 | 1.56 | 1.65 | 1.74 | 1.85 | 1.97 | 2.1 | 2.24 | 2.4 | 0.0 | 0.0 | 0.0 | 0.0 | 0.0 | 0.0 | 0.0 | 0.0 | 0.0 | 0.0 | 0.0 | 0.0 | 0.0 | 0.0 | 0.0 | 0.0 |
| 0.76 | 1.26 | 1.33 | 1.4 | 1.47 | 1.56 | 1.65 | 1.74 | 1.85 | 1.97 | 2.1 | 2.24 | 2.4 | 2.57 | 0.0 | 0.0 | 0.0 | 0.0 | 0.0 | 0.0 | 0.0 | 0.0 | 0.0 | 0.0 | 0.0 | 0.0 | 0.0 | 0.0 | 0.0 |
| 0.74 | 1.26 | 1.33 | 1.4 | 1.47 | 1.56 | 1.65 | 1.74 | 1.85 | 1.97 | 2.1 | 2.24 | 2.4 | 2.57 | 2.76 | 0.0 | 0.0 | 0.0 | 0.0 | 0.0 | 0.0 | 0.0 | 0.0 | 0.0 | 0.0 | 0.0 | 0.0 | 0.0 | 0.0 |
| 0.71 | 1.26 | 1.33 | 1.4 | 1.47 | 1.56 | 1.65 | 1.74 | 1.85 | 1.97 | 2.1 | 2.24 | 2.4 | 2.57 | 2.76 | 2.98 | 0.0 | 0.0 | 0.0 | 0.0 | 0.0 | 0.0 | 0.0 | 0.0 | 0.0 | 0.0 | 0.0 | 0.0 | 0.0 |
| 0.68 | 1.26 | 1.33 | 1.4 | 1.47 | 1.56 | 1.65 | 1.74 | 1.85 | 1.97 | 2.1 | 2.24 | 2.4 | 2.57 | 2.76 | 2.98 | 3.22 | 0.0 | 0.0 | 0.0 | 0.0 | 0.0 | 0.0 | 0.0 | 0.0 | 0.0 | 0.0 | 0.0 | 0.0 |
| 0.64 | 1.26 | 1.33 | 1.4 | 1.47 | 1.56 | 1.65 | 1.74 | 1.85 | 1.97 | 2.1 | 2.24 | 2.4 | 2.57 | 2.76 | 2.98 | 3.22 | 3.49 | 0.0 | 0.0 | 0.0 | 0.0 | 0.0 | 0.0 | 0.0 | 0.0 | 0.0 | 0.0 | 0.0 |
| 0.60 | 1.26 | 1.33 | 1.4 | 1.47 | 1.56 | 1.65 | 1.74 | 1.85 | 1.97 | 2.1 | 2.24 | 2.4 | 2.57 | 2.76 | 2.98 | 3.22 | 3.49 | 3.8 | 0.0 | 0.0 | 0.0 | 0.0 | 0.0 | 0.0 | 0.0 | 0.0 | 0.0 | 0.0 |
| 0.56 | 1.26 | 1.33 | 1.4 | 1.47 | 1.56 | 1.65 | 1.74 | 1.85 | 1.97 | 2.1 | 2.24 | 2.4 | 2.57 | 2.76 | 2.98 | 3.22 | 3.49 | 3.8 | 4.15 | 0.0 | 0.0 | 0.0 | 0.0 | 0.0 | 0.0 | 0.0 | 0.0 | 0.0 |
| 0.52 | 1.26 | 1.33 | 1.4 | 1.47 | 1.56 | 1.65 | 1.74 | 1.85 | 1.97 | 2.1 | 2.24 | 2.4 | 2.57 | 2.76 | 2.98 | 3.22 | 3.49 | 3.8 | 4.15 | 4.56 | 0.0 | 0.0 | 0.0 | 0.0 | 0.0 | 0.0 | 0.0 | 0.0 |
| 0.46 | 1.26 | 1.33 | 1.4 | 1.47 | 1.56 | 1.65 | 1.74 | 1.85 | 1.97 | 2.1 | 2.24 | 2.4 | 2.57 | 2.76 | 2.98 | 3.22 | 3.49 | 3.8 | 4.15 | 4.56 | 5.02 | 0.0 | 0.0 | 0.0 | 0.0 | 0.0 | 0.0 | 0.0 |
| 0.41 | 1.26 | 1.33 | 1.4 | 1.47 | 1.56 | 1.65 | 1.74 | 1.85 | 1.97 | 2.1 | 2.24 | 2.4 | 2.57 | 2.76 | 2.98 | 3.22 | 3.49 | 3.8 | 4.15 | 4.56 | 5.02 | 5.56 | 0.0 | 0.0 | 0.0 | 0.0 | 0.0 | 0.0 |
| 0.35 | 1.26 | 1.33 | 1.4 | 1.47 | 1.56 | 1.65 | 1.74 | 1.85 | 1.97 | 2.1 | 2.24 | 2.4 | 2.57 | 2.76 | 2.98 | 3.22 | 3.49 | 3.8 | 4.15 | 4.56 | 5.02 | 5.56 | 6.19 | 0.0 | 0.0 | 0.0 | 0.0 | 0.0 |
| 0.28 | 1.26 | 1.33 | 1.4 | 1.47 | 1.56 | 1.65 | 1.74 | 1.85 | 1.97 | 2.1 | 2.24 | 2.4 | 2.57 | 2.76 | 2.98 | 3.22 | 3.49 | 3.8 | 4.15 | 4.56 | 5.02 | 5.56 | 6.19 | 6.93 | 0.0 | 0.0 | 0.0 | 0.0 |
| 0.20 | 1.26 | 1.33 | 1.4 | 1.47 | 1.56 | 1.65 | 1.74 | 1.85 | 1.97 | 2.1 | 2.24 | 2.4 | 2.57 | 2.76 | 2.98 | 3.22 | 3.49 | 3.8 | 4.15 | 4.56 | 5.02 | 5.56 | 6.19 | 6.93 | 7.82 | 0.0 | 0.0 | 0.0 |
| 0.11 | 1.26 | 1.33 | 1.4 | 1.47 | 1.56 | 1.65 | 1.74 | 1.85 | 1.97 | 2.1 | 2.24 | 2.4 | 2.57 | 2.76 | 2.98 | 3.22 | 3.49 | 3.8 | 4.15 | 4.56 | 5.02 | 5.56 | 6.19 | 6.93 | 7.82 | 8.89 | 0.0 | 0.0 |
| 0.01 | 1.26 | 1.33 | 1.4 | 1.47 | 1.56 | 1.65 | 1.74 | 1.85 | 1.97 | 2.1 | 2.24 | 2.4 | 2.57 | 2.76 | 2.98 | 3.22 | 3.49 | 3.8 | 4.15 | 4.56 | 5.02 | 5.56 | 6.19 | 6.93 | 7.82 | 8.89 | 10.2 | 0.0 |
| 0.00 | 1.26 | 1.33 | 1.4 | 1.47 | 1.56 | 1.65 | 1.74 | 1.85 | 1.97 | 2.1 | 2.24 | 2.4 | 2.57 | 2.76 | 2.98 | 3.22 | 3.49 | 3.8 | 4.15 | 4.56 | 5.02 | 5.56 | 6.19 | 6.93 | 7.82 | 8.89 | 10.2 | 0.89 |

For example, the first row should be normalized to 0.0126, and the second row should be normalized to 0.0259. The usage of Table 13 is the same.

Table 13: SD3's signal coefficient matrix with more sharpness

| time | 1.00 | 0.99 | 0.97 | 0.96 | 0.95 | 0.93 | 0.91 | 0.90 | 0.88 | 0.86 | 0.84 | 0.81 | 0.79 | 0.76 | 0.74 | 0.71 | 0.68 | 0.64 | 0.60 | 0.56 | 0.52 | 0.46 | 0.41 | 0.35 | 0.28 | 0.20 | 0.11 | 0.01 |
|---|---|---|---|---|---|---|---|---|---|---|---|---|---|---|---|---|---|---|---|---|---|---|---|---|---|---|---|---|
| 0.99 | 1.26 | 0.0 | 0.0 | 0.0 | 0.0 | 0.0 | 0.0 | 0.0 | 0.0 | 0.0 | 0.0 | 0.0 | 0.0 | 0.0 | 0.0 | 0.0 | 0.0 | 0.0 | 0.0 | 0.0 | 0.0 | 0.0 | 0.0 | 0.0 | 0.0 | 0.0 | 0.0 | 0 |
| 0.97 | 1.26 | 1.33 | 0.0 | 0.0 | 0.0 | 0.0 | 0.0 | 0.0 | 0.0 | 0.0 | 0.0 | 0.0 | 0.0 | 0.0 | 0.0 | 0.0 | 0.0 | 0.0 | 0.0 | 0.0 | 0.0 | 0.0 | 0.0 | 0.0 | 0.0 | 0.0 | 0.0 | 0 |
| 0.96 | 1.26 | 1.33 | 1.4 | 0.0 | 0.0 | 0.0 | 0.0 | 0.0 | 0.0 | 0.0 | 0.0 | 0.0 | 0.0 | 0.0 | 0.0 | 0.0 | 0.0 | 0.0 | 0.0 | 0.0 | 0.0 | 0.0 | 0.0 | 0.0 | 0.0 | 0.0 | 0.0 | 0 |
| 0.95 | 0.0 | 1.33 | 1.4 | 1.47 | 0.0 | 0.0 | 0.0 | 0.0 | 0.0 | 0.0 | 0.0 | 0.0 | 0.0 | 0.0 | 0.0 | 0.0 | 0.0 | 0.0 | 0.0 | 0.0 | 0.0 | 0.0 | 0.0 | 0.0 | 0.0 | 0.0 | 0.0 | 0 |
| 0.93 | 0.0 | 1.33 | 1.4 | 1.47 | 1.56 | 0.0 | 0.0 | 0.0 | 0.0 | 0.0 | 0.0 | 0.0 | 0.0 | 0.0 | 0.0 | 0.0 | 0.0 | 0.0 | 0.0 | 0.0 | 0.0 | 0.0 | 0.0 | 0.0 | 0.0 | 0.0 | 0.0 | 0 |
| 0.91 | 0.0 | 0.0 | 1.44 | 1.56 | 1.56 | 1.65 | 0.0 | 0.0 | 0.0 | 0.0 | 0.0 | 0.0 | 0.0 | 0.0 | 0.0 | 0.0 | 0.0 | 0.0 | 0.0 | 0.0 | 0.0 | 0.0 | 0.0 | 0.0 | 0.0 | 0.0 | 0.0 | 0 |
| 0.90 | 0.0 | 0.0 | 0.0 | 0.0 | 1.56 | 1.65 | 1.74 | 0.0 | 0.0 | 0.0 | 0.0 | 0.0 | 0.0 | 0.0 | 0.0 | 0.0 | 0.0 | 0.0 | 0.0 | 0.0 | 0.0 | 0.0 | 0.0 | 0.0 | 0.0 | 0.0 | 0.0 | 0 |
| 0.88 | 0.0 | 0.0 | 0.0 | 0.0 | 0.0 | 1.65 | 1.74 | 1.85 | 0.0 | 0.0 | 0.0 | 0.0 | 0.0 | 0.0 | 0.0 | 0.0 | 0.0 | 0.0 | 0.0 | 0.0 | 0.0 | 0.0 | 0.0 | 0.0 | 0.0 | 0.0 | 0.0 | 0 |
| 0.86 | 0.0 | 0.0 | 0.0 | 0.0 | 0.0 | 1.65 | 1.74 | 1.85 | 1.97 | 0.0 | 0.0 | 0.0 | 0.0 | 0.0 | 0.0 | 0.0 | 0.0 | 0.0 | 0.0 | 0.0 | 0.0 | 0.0 | 0.0 | 0.0 | 0.0 | 0.0 | 0.0 | 0 |
| 0.84 | 0.0 | 0.0 | 0.0 | 0.0 | 0.0 | 0.0 | 1.74 | 1.85 | 1.97 | 2.1 | 0.0 | 0.0 | 0.0 | 0.0 | 0.0 | 0.0 | 0.0 | 0.0 | 0.0 | 0.0 | 0.0 | 0.0 | 0.0 | 0.0 | 0.0 | 0.0 | 0.0 | 0 |
| 0.81 | 0.0 | 0.0 | 0.0 | 0.0 | 0.0 | 0.0 | 0.0 | 1.85 | 1.97 | 2.1 | 2.24 | 0.0 | 0.0 | 0.0 | 0.0 | 0.0 | 0.0 | 0.0 | 0.0 | 0.0 | 0.0 | 0.0 | 0.0 | 0.0 | 0.0 | 0.0 | 0.0 | 0 |
| 0.79 | 0.0 | 0.0 | 0.0 | 0.0 | 0.0 | 0.0 | 0.0 | 0.0 | 1.97 | 2.1 | 2.24 | 2.4 | 0.0 | 0.0 | 0.0 | 0.0 | 0.0 | 0.0 | 0.0 | 0.0 | 0.0 | 0.0 | 0.0 | 0.0 | 0.0 | 0.0 | 0.0 | 0 |
| 0.76 | 0.0 | 0.0 | 0.0 | 0.0 | 0.0 | 0.0 | 0.0 | 0.0 | 0.0 | 2.1 | 2.24 | 2.4 | 2.57 | 0.0 | 0.0 | 0.0 | 0.0 | 0.0 | 0.0 | 0.0 | 0.0 | 0.0 | 0.0 | 0.0 | 0.0 | 0.0 | 0.0 | 0 |
| 0.74 | 0.0 | 0.0 | 0.0 | 0.0 | 0.0 | 0.0 | 0.0 | 0.0 | 0.0 | 2.1 | 2.24 | 2.4 | 2.57 | 2.76 | 0.0 | 0.0 | 0.0 | 0.0 | 0.0 | 0.0 | 0.0 | 0.0 | 0.0 | 0.0 | 0.0 | 0.0 | 0.0 | 0 |
| 0.71 | 0.0 | 0.0 | 0.0 | 0.0 | 0.0 | 0.0 | 0.0 | 0.0 | 0.0 | 2.1 | 2.24 | 2.4 | 2.57 | 2.76 | 2.98 | 0.0 | 0.0 | 0.0 | 0.0 | 0.0 | 0.0 | 0.0 | 0.0 | 0.0 | 0.0 | 0.0 | 0.0 | 0 |
| 0.68 | 0.0 | 0.0 | 0.0 | 0.0 | 0.0 | 0.0 | 0.0 | 0.0 | 0.0 | 2.1 | 2.24 | 2.4 | 2.57 | 2.76 | 2.98 | 3.22 | 0.0 | 0.0 | 0.0 | 0.0 | 0.0 | 0.0 | 0.0 | 0.0 | 0.0 | 0.0 | 0.0 | 0 |
| 0.64 | 0.0 | 0.0 | 0.0 | 0.0 | 0.0 | 0.0 | 0.0 | 0.0 | 0.0 | 2.1 | 2.24 | 2.4 | 2.57 | 2.76 | 2.98 | 3.22 | 3.49 | 0.0 | 0.0 | 0.0 | 0.0 | 0.0 | 0.0 | 0.0 | 0.0 | 0.0 | 0.0 | 0 |
| 0.60 | 0.0 | 0.0 | 0.0 | 0.0 | 0.0 | 0.0 | 0.0 | 0.0 | 0.0 | 2.1 | 2.24 | 2.4 | 2.57 | 2.76 | 2.98 | 3.22 | 3.49 | 3.8 | 0.0 | 0.0 | 0.0 | 0.0 | 0.0 | 0.0 | 0.0 | 0.0 | 0.0 | 0 |
| 0.56 | 0.0 | 0.0 | 0.0 | 0.0 | 0.0 | 0.0 | 0.0 | 0.0 | 0.0 | 0.0 | 2.24 | 2.4 | 2.57 | 2.76 | 2.98 | 3.22 | 3.49 | 3.8 | 4.15 | 0.0 | 0.0 | 0.0 | 0.0 | 0.0 | 0.0 | 0.0 | 0.0 | 0 |
| 0.52 | 0.0 | 0.0 | 0.0 | 0.0 | 0.0 | 0.0 | 0.0 | 0.0 | 0.0 | 0.0 | 2.24 | 2.4 | 2.57 | 2.76 | 2.98 | 3.22 | 3.49 | 3.8 | 4.15 | 4.56 | 0.0 | 0.0 | 0.0 | 0.0 | 0.0 | 0.0 | 0.0 | 0 |
| 0.46 | 0.0 | 0.0 | 0.0 | 0.0 | 0.0 | 0.0 | 0.0 | 0.0 | 0.0 | 0.0 | 2.24 | 2.4 | 2.57 | 2.76 | 2.98 | 3.22 | 3.49 | 3.8 | 4.15 | 4.56 | 5.02 | 0.0 | 0.0 | 0.0 | 0.0 | 0.0 | 0.0 | 0 |
| 0.41 | 0.0 | 0.0 | 0.0 | 0.0 | 0.0 | 0.0 | 0.0 | 0.0 | 0.0 | 0.0 | 2.24 | 2.4 | 2.57 | 2.76 | 2.98 | 3.22 | 3.49 | 3.8 | 4.15 | 4.56 | 5.02 | 5.56 | 0.0 | 0.0 | 0.0 | 0.0 | 0.0 | 0 |
| 0.35 | 0.0 | 0.0 | 0.0 | 0.0 | 0.0 | 0.0 | 0.0 | 0.0 | 0.0 | 0.0 | 0.0 | 0.0 | 2.57 | 2.76 | 2.98 | 3.22 | 3.49 | 3.8 | 4.15 | 4.56 | 5.02 | 5.56 | 6.19 | 0.0 | 0.0 | 0.0 | 0.0 | 0 |
| 0.28 | 0.0 | 0.0 | 0.0 | 0.0 | 0.0 | 0.0 | 0.0 | 0.0 | 0.0 | 0.0 | 0.0 | 0.0 | 2.57 | 2.76 | 2.98 | 3.22 | 3.49 | 3.8 | 4.15 | 4.56 | 5.02 | 5.56 | 6.19 | 6.93 | 0.0 | 0.0 | 0.0 | 0 |
| 0.20 | 0.0 | 0.0 | 0.0 | 0.0 | 0.0 | 0.0 | 0.0 | 0.0 | 0.0 | 0.0 | 0.0 | 0.0 | 2.57 | 2.76 | 2.98 | 3.22 | 3.49 | 3.8 | 4.15 | 4.56 | 5.02 | 5.56 | 6.19 | 6.93 | 7.82 | 0.0 | 0.0 | 0 |
| 0.11 | 0.0 | 0.0 | 0.0 | 0.0 | 0.0 | 0.0 | 0.0 | 0.0 | 0.0 | 0.0 | 0.0 | 0.0 | 2.57 | 2.76 | 2.98 | 3.22 | 3.49 | 3.8 | 4.15 | 4.56 | 5.02 | 5.56 | 6.19 | 6.93 | 7.82 | 8.89 | 0.0 | 0 |
| 0.01 | 0.0 | 0.0 | 0.0 | 0.0 | 0.0 | 0.0 | 0.0 | 0.0 | 0.0 | 0.0 | 0.0 | 0.0 | 0.0 | 0.0 | 2.98 | 3.22 | 3.49 | 3.8 | 4.15 | 4.56 | 5.02 | 5.56 | 6.19 | 6.93 | 7.82 | 8.89 | 10.2 | 0 |
| 0.00 | 0.0 | 0.0 | 0.0 | 0.0 | 0.0 | 0.0 | 0.0 | 0.0 | 0.0 | 0.0 | 0.0 | 0.0 | 0.0 | 0.0 | 2.98 | 3.22 | 3.49 | 3.8 | 4.15 | 4.56 | 5.02 | 5.56 | 6.19 | 6.93 | 7.82 | 8.89 | 10.2 | 15 |

### E.2 INFERENCE PROCESS VISUALIZATION

Figures 15 and 16 provide a visualization of the complete inference process. The left half shows the inference process using the coefficient matrix from Table 12, and the right half shows the inference process using the coefficient matrix from Table 13. The first column shows the result of Self Guidance, which is also the input image signal to the model. The second column shows the model output without conditioning, the third column shows the conditioned model output, and the fourth column shows the result of Classifier Free Guidance. For each model operation, there is a clear image signal input and image signal output, which greatly enhances intuitive understanding of the operation's purpose and facilitates efficient debugging and problem analysis.

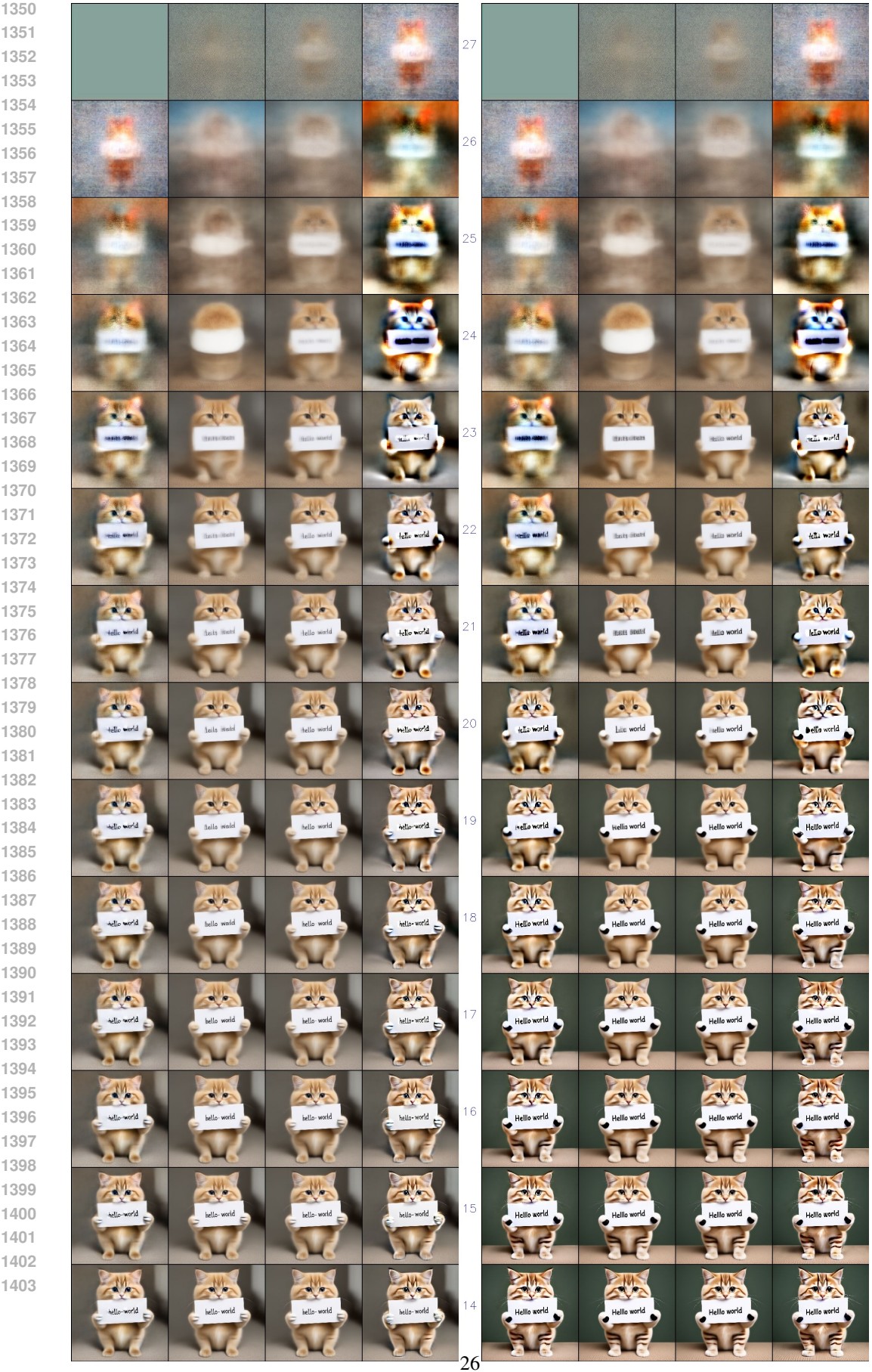

Figure 15: Inference process visualization: first half.

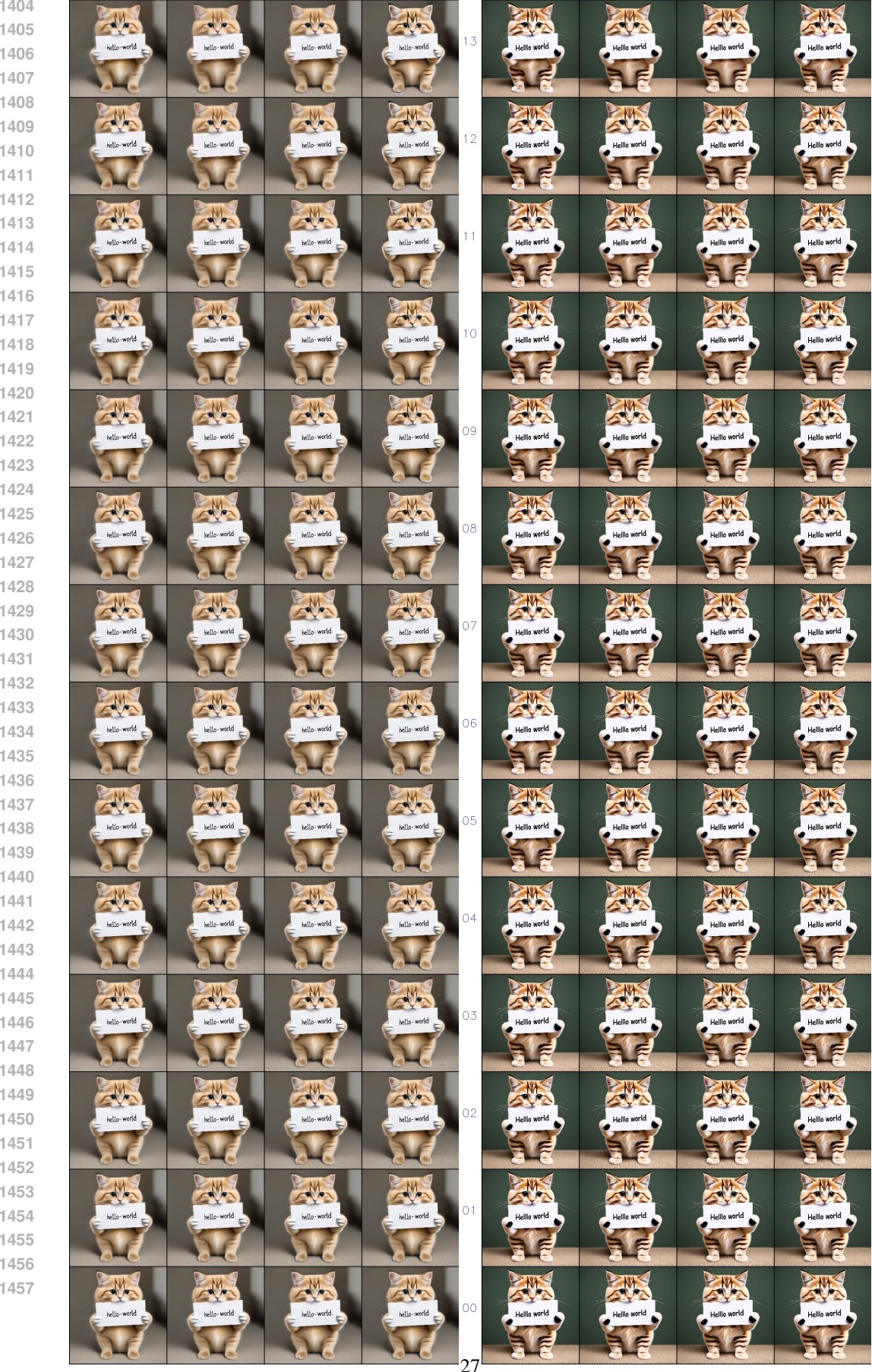

Figure 16: Inference process visualization: second half