# OpenReview forum: "Rethinking Diffusion Model in High Dimension"
_ICLR.cc/2026/Conference — Submitted to ICLR 2026_

### Official Review · Reviewer_k5VA · 2025-10-21

**Soundness:** 1
**Presentation:** 1
**Contribution:** 1
**Rating:** 0
**Confidence:** 5

**Summary:**

The paper analyzes the diffusion and flow matching models and argues that the learning objective suffers from sparsity in higher dimensions.

**Strengths:**

I think that the problem of analyzing diffusion/flow models' objective is an important one.

**Weaknesses:**

The papers has the following strong weaknesses:
1. **Lack of proper positioning/context in the literature**. The authors make a very strong and false claim in the introduction that they "present the first rigorous analysis of the diffusion model objective in high-dimensional sparse scenario". This problem is a known one and an active area of research. See for example [1] and references therein. The authors do not acknowledge, cite or compare with the existing literature on the topic. Furthermore, a large portion of the proofs (e.g. appendix A) are known results and the authors do not explicitly say that, potentially misleading readers that these are new results. Similarly, the entire discussion in section 2 about the similarities between diffusion and flow matching is well known, and nicely explained e.g. in the blog post: https://diffusionflow.github.io/. The authors should be more explicit that all of this is already known and properly cite related works.
2. **The paper lacks mathematical rigour**. A lot of statements are vague and sometimes I genuinely did not know what the authors were trying to convey. For example
    1. what does 'variance is relatively fixed' mean?
    2. The notation is also confusing. What does E
    3. What does "Furthermore, due to limited sampling during training, each $p(x0|xt = Xt)$ cannot be sufficiently sampled, so the actual degradation ratio should be higher than the statistics show" mean?
    4. What does "In high dimensions, each $p(x0|xt = Xt)$ should be complex" mean?
    5. "It easily predicts non-submerged frequencies (likely copying them)." What are non-submerged frequencies?
    6. "Since the model compensates for the submerged frequencies, it can also be regarded as an information enhancement operator." - I don't understand what this sentence means.
    7. "Based on the principle of train-test matching". What is the principle of train-test matching?
    8. The entire section 4.1 lacks mathematical rigour. I_good and I_bad should be explicitly defined.
    9. "The linear combination of independent noise is still noise"
3. **There is no contribution**. Finally, the paper presents no contribution. They argue that the model objective is suboptimal. What they propose instead is "Self guidance", which in principle sounds similar to "Autoguidance" [2] (which is not cited) and "natural inference", but the authors provide no details on how it actually works. The entire section 4.2 is tough to understand. Figure 5 makes it more confusing rather than clarifying. Instead of rigorously defining what the authors propose with equations, they provide a list of bullet points that summarize the "core ideas". Finally, the authors do not provide any empirical results with their new framework, and how they compare with known sampling frameworks.

Some more details:

1. The proofs in appendix A:
    1. Are imprecise - f_\theta is not defined. I assume that it's a function from R^D to R^D. But there are typos with missing norms, i.e. f_\theta^2 is undefined when f is not one-dimensional
    2. All these results are well known are derived multiple times in various works. The authors don't mention this anywhere, and might mislead someone into thinking that these are novel results
2. The discussion about flow matching in lines 138-152 is not correct in general. A flow matching model can be defined with various probability paths (and in particular, the prior distribution need not be Gaussian). It should be mentioned here explicitly that the authors only mean this simplified case (which is equivalent to a diffusion model with a certain noise schedule)
3. Line 231. The reason for flow matching showing different statistics is that it uses a different noise schedule than VP. In other words, the "time" $t$ means different levels of noise for both models.
4. "When weighted sum degradation occurs, it is equivalent to using a single sample as an estimator of the mean, which typically have large error". I have two issues with this statement
    1. it is not true. The authors define degradation as the probability distribution over training data having an entry > 0.9. This is not equivalent to a dirac delta.
    2. Secondly, we do the same thing with VAEs - we learn the model by only using a single sample, and it works fine in practice.
5. "If we cannot provide an accurate fitting target, we argue that the model is unlikely to learn the ideal target accurately"
    1. The question is: do we want to lean the ideal target? The ideal target will never generate any unseen data, because of the form of Eq14. So it is not really desirable to try to learn the ideal target.
    2. Therefore, in practive, diffusion models certainly do not learn the ideal target, and it's good, because they are able to generate unseen examples. An interesting question is why that happens.
6. Figure 5 is very difficult to parse. Some undefined symbols appear: a_t, b_t, c_t, y_t, \hat{a}_t.

---
References:

[1] Lukoianov et al. "Locality in Image Diffusion Models Emerges from Data Statistics" (NeurIPS 2025)

[2] Karras et al. "Guiding a Diffusion Model with a Bad Version of Itself" (NeurIPS 2024)

**Questions:**

1. In tables 1 and 2, for time=600, it seems like the "degradation" is much more pronounced than "degradation to x0". This is not intuitive to me. Are you saying that after adding some amount of noise I am much more likely to be closer to a different training example than the one I started with, and the weighted sum is still degenerate?
2. "When training a model to predict X0 from noise-mixed samples, the model prioritizes frequencies based on their SNR"
    1. Is this claim proven anywhere in the paper?
3. "This frequency-dependent process is confirmed during inference: early steps (large t) generate contours, while later steps (small t) add details."
    1. I think this is indeed considered general knowledge in the field, but I don't see the authors providing evidence of this claim, nor citing any works that demonstrate this

---

> ### Author Response · Authors · 2025-11-24
>
> We sincerely thank the reviewer for the numerous comments and questions. Reviewing a theoretically oriented paper requires substantial time and effort, and we truly appreciate it. Given the large number of questions, we prioritize addressing some of the key and core issues. We hope that our discussion will help deepen the understanding of diffusion models.
>
> $&nbsp;$
>
>
> $\textbf{Comment}$:
> > #Lack of proper positioning/context in the literature#. The authors make a very strong and false claim in the introduction that they "present the first rigorous analysis of the diffusion model objective in high-dimensional sparse scenario". This problem is a known one and an active area of research. See for example [1] and references therein. The authors do not acknowledge, cite or compare with the existing literature on the topic. .....
>
> $\textbf{Answer}$ :
> The article [1] you mentioned is indeed insightful, but it was first made publicly available in September 2025. Our paper was initially published in March 2025, so we were not able to cite it in time.
>
> In addition, the related works you referred to mainly discuss issues concerning ideal denoisers (i.e., the closed-form score of the empirical distribution), and they do not explicitly analyze how high-dimensional sparsity affects the objective function. Therefore, we believe that our claim—“present the first rigorous analysis of the diffusion model objective in the high-dimensional sparse scenario”—is appropriate. Of course, these works are all valuable, and we will include citations to them in the revised version.
>
> The proof in Appendix A serves as a supplementary explanation to Section 2 – Background. It summarizes existing information only to prepare for the later discussion on the impact of sparsity, and it does not claim to be a contribution of this paper.
>
> $&nbsp;$
>
> $\textbf{Question}$:
> > "If we cannot provide an accurate fitting target, we argue that the model is unlikely to learn the ideal target accurately"
> The question is: do we want to lean the ideal target? The ideal target will never generate any unseen data, because of the form of Eq14. So it is not really desirable to try to learn the ideal target.
> Therefore, in practive, diffusion models certainly do not learn the ideal target, and it's good, because they are able to generate unseen examples. An interesting question is why that happens.
>
> $\textbf{Answer}$ :
> In the phrase “learn the ideal target accurately,” the ideal target refers to the score of the underlying data distribution, rather than the score of the empirical distribution. This differs somewhat from your understanding. You mentioned that the ideal target never generates new samples; this is because the score corresponding to the empirical distribution (or the mean of $p(x_0 \mid x_t)$) is inaccurate and degenerates to a single training sample $x_0$, which in turn leads to generating only samples from the training set.
>
> Moreover, here we do not emphasize that one must learn the exact score of the underlying data distribution (or the mean of $p(x_0 \mid x_t)$). What we aim to illustrate is that, due to high-dimensional sparsity, the score of the empirical distribution (or the mean of $p(x_0 \mid x_t)$) differs significantly from the score of the true data distribution (or the mean of $p(x_0 \mid x_t)$). Therefore, when a neural network is used to fit the score of the empirical distribution, it cannot learn the true data distribution’s score; instead, it captures other information.
>
> Additionally, we also show that due to sparsity, the mean of $p(x_0 \mid x_t)$ corresponding to the empirical distribution degenerates to a single $x_0$. Hence, we can provide an alternative perspective to explain this degenerated objective function: from a spectral viewpoint, predicting $x_0$ from $x_t$ can be interpreted as adding new frequency components. Furthermore, we unify existing inference methods into a new form, which aligns with the degenerated objective function and consists of a series of operations predicting $x_0$ from $x_t$, effectively corresponding to a series of operations that add new frequency components. This naturally explains why diffusion models are able to generate new samples.
>
> $&nbsp;$
>
> $\textbf{Comment}$:
> > There is no contribution . Finally, the paper presents no contribution. They argue that the model objective is suboptimal. What they propose instead is "Self guidance", which in principle sounds similar to "Autoguidance" [2] (which is not cited) and "natural inference", ......
>
> $\textbf{Answer}$ :
> This paper does not claim that "the model objective is suboptimal", nor does it propose new inference methods. Its main purpose is to provide a completely new and intuitive perspective for understanding the existing objective functions and inference methods. This perspective does not rely on any statistical concepts and can explain why the inference methods of diffusion models are capable of generating new samples.

---

> ### Author Response · Authors · 2025-11-25
> **Second Response**
>
> $&nbsp;$
>
> $\textbf{Question 1}$:
> >In tables 1 and 2, for time=600, it seems like the "degradation" is much more pronounced than "degradation to x0". This is not intuitive to me. Are you saying that after adding some amount of noise I am much more likely to be closer to a different training example than the one I started with, and the weighted sum is still degenerate?
>
> $\textbf{Answer}$ :
> Yes, this is indeed possible when the noise level is large. For example, consider two points $A$ and $B$ in a 2D space, where$A=(0,0),\ B=(0,5)$, and the distance between them is 5. After adding sufficiently large noise, suppose $A$ is shifted to a new position $A^\prime=(0,4)$. n this case, the distance$A^\prime A=4, A^\prime B = 1$. Under such circumstances, $p(B \mid A^\prime)$ becomes significantly larger than $p(A \mid A^\prime)$, meaning that the sample may indeed degenerate toward another data point in the dataset.
>
> The above describes the exact degeneration ratio under the empirical distribution. However, in practical training, another factor also influences the proportion of degeneration toward $x_0$. Specifically, during training we do not excatly compute the true empirical mean of $p(x_0 \mid x_t)$. Instead, we approximate the mean via Monte Carlo estimation, where sampling from $p(x_0 \mid x_t)$ is implemented using Ancestral Sampling. In high-dimensional sparse spaces, Ancestral Sampling rarely provides multiple samples for a specific conditional distribution $p(x_0 \mid x_t = x_t')$, which leads to a higher chance of degeneration toward the specific $x_0$ used in that sample. The reasoning is as follows:
>
> During training, we approximate the mean of $p(x_0 \mid x_t)$ using Monte Carlo estimation, whose essential step is sampling from $p(x_0 \mid x_t)$. This is implemented via Ancestral Sampling. Concretely, we first randomly select a sample $x_0$ from the dataset (equivalent to sampling from the empirical distribution $p_{\text{empirical}}(x) = \frac{1}{N-1} \sum_{i=0}^N \delta(x - x_0^i)$, denoted $x_0'$. We then apply the noise-addition process to obtain $x_t$ (equivalent to sampling from $p(x_t \mid x_0)$), denoted $x_t'$. This produces the pair $(x_0', x_t') \sim p(x_0, x_t)$, and in particular $x_0' \sim p(x_0 \mid x_t = x_t')$. Repeating this process yields multiple pairs: $(x_0', x_t')^0, (x_0', x_t')^1, \dots, (x_0', x_t')^K$.
>
> While this procedure indeed samples from $p(x_0 \mid x_t = x_t')$, it does not generate multiple samples for the same condition $x_t'$. This is because, as dimensionality increases, the space becomes extremely large, making it unlikely for two sampled $x_t'$ values to be similar or identical. As a result, the mean of each conditional distribution $p(x_0 \mid x_t)$ is approximated using only one sample — namely, that specific $x_0'$. Therefore, during training, degeneration toward $x_0$ occurs more frequently.

---

> ### Author Response · Authors · 2025-11-25
> **Third Response**
>
> $&nbsp;$
>
> $\textbf{Question 2}$:
> >"When training a model to predict X0 from noise-mixed samples, the model prioritizes frequencies based on their SNR"
> Is this claim proven anywhere in the paper?
>
> $\textbf{Answer}$ :
> This viewpoint is the main focus of Section~3.3. Here, we provide a further explanation.
>
> The degraded objective is to predict $x_0$ from $x_t$, where $x_t = t \cdot \varepsilon + (1 - t) \cdot x_0$
>
> First, we know that noise has a \textbf{uniform frequency spectrum}, meaning all frequency components have equal amplitude. In contrast, real images have a \textbf{non-uniform spectrum}: low-frequency components have large amplitudes, and high-frequency components have small amplitudes. Therefore, when noise is mixed with the image signal, the \textbf{SNR of high-frequency components is always lower} than that of low-frequency components.
>
> Additionally, during optimization we use the \textbf{Euclidean loss}. This loss can be decomposed in the frequency domain into multiple Euclidean losses, each corresponding to a small band of frequencies. Because low-frequency components have large amplitudes (which means their corresponding losses have larger weights) and also have high SNR, the optimization process naturally prioritizes fitting low frequencies---the lower the frequency, the higher the priority. Moreover, each frequency band has a different prediction difficulty: frequencies with high SNR (not submerged by noise) are easier for the model to predict---can simply copying them from the input. Frequencies with low SNR (submerged by noise) are much harder to predict.
>
> At $\textbf{large $t$}$, noise is strong, and the SNR of most frequency components---including low frequencies---is low. These are all difficult-to-predict components. As noted above, the model prioritizes low-frequency components, so it mainly focuses on predicting low frequencies and does not have the capacity to handle high-frequency components.
>
> At $\textbf{small $t$}$, noise is weak, and low-frequency components have relatively high SNR. These components are easy to predict, so the model quickly reconstructs them and can then allocate more capacity to predict the more challenging high-frequency components.
>
> $&nbsp;$
>
> $\textbf{Question 3}$:
> >"This frequency-dependent process is confirmed during inference: early steps (large t) generate contours, while later steps (small t) add details."
> I think this is indeed considered general knowledge in the field, but I don't see the authors providing evidence of this claim, nor citing any works that demonstrate this
>
> $\textbf{Answer}$ :
> See the answer to the previous question.

---

### Official Review · Reviewer_nswE · 2025-10-29

**Soundness:** 1
**Presentation:** 1
**Contribution:** 1
**Rating:** 2
**Confidence:** 3

**Summary:**

This article posits that diffusion models are fundamentally limited in their ability to learn high-dimensional distributions with sparsity. To address this, the authors introduce a natural inference process that serves to unify existing methodologies.

**Strengths:**

- **Significance**: This article concerns the problem of diffusion models where the data distribution is sparse in high dimension, which could be a critical problem in the real-world applications.

**Weaknesses:**

- **Soundness**:
  - Unknown facts: The authors demonstrate that in high-dimensional sparse scenarios, machine learning models can not effectively learn complex hidden probability distributions and their essential statistical quantities, but they do not show the references or provide other evidence.
  - Equation (15) is unclear, since the expectation is taken over $x_{t}$ rather than $x_{0}$.
  - Figure 1 seems to be incorrect, since $x_{t}$ can be at positions near other data points other than $x_{0}$.
- **Completeness**:
  - Lack of experiments in the major text.
  - Lack of a related work section that contains the relevent references of this study.
- **Reresentation**:
  - Redundancy: (1) In Sec. 1, the descriptions of assumptions made for diffusion models, flow-matching models; (2) the proofs for the equivalence of score-matching objective and denoising score matching objective in Sec. A.1 is not necessary, since it is well-known and not made by the authors.
  - Formatting Distractions: The overuse of emphasized text (e.g., excessive bolding or italics) biases the reader's attention and disrupts the flow of the narrative.
  - Inproper figures: (1) Figure 5 is overly cluttered with mathematical symbols and equations that are not essential to its point and are not adequately explained in the caption or main text.; (2) all figure captions are missing a period at the end.
  - In Sec. 3.1, the text reads "Eq. equation (1)" instead of the standard "Eq. (1)". This formatting error should be corrected throughout the manuscript.

**Questions:**

- How can this study be applied in the real-world applications?

---

> ### Author Response · Authors · 2025-11-22
> **First Response**
>
> We sincerely thank the reviewer for the careful and thoughtful comments. Reviewing a theoretically oriented paper requires substantial time and effort, which we truly appreciate. Below, we provide  responses to the key comments , and we hope that our discussion can help deepen the understanding of diffusion models.
>
> $&nbsp;$
>
> $\textbf{comment 1}$:
> >Unknown facts: The authors demonstrate that in high-dimensional setting, machine learning models can not effectively learn complex hidden probability distributions and their essential statistical quantities, but they do not show the references or provide other evidence.
>
> $\textbf{Response}$:
> This article justifies this claim as follows.
>
> First, in Section~2 (Background), we establish that the essential objective of diffusion model training is to fit the mean of the posterior distribution $p(x_0 \mid x_t)$, namely $\mathbb{E}_{p(x_0 \mid x_t)}[x_0]$. This mean can be interpreted as a weighted combination of multiple $x_0$ samples, and various statistical quantities—including the score and velocity—can be derived directly from it.
>
> Second, in Section-3, we show that in high dimension spaces, due to data sparsity, replacing the true data distribution with the empirical distribution (a replacement that always occurs when sampling from $p_{data}(x_0)$ during training) causes $\mathbb{E}_{p(x_0 \mid x_t)}[x_0]$ to degenerate into a single $x_0$ sample. We argue that this degenerate mean deviates significantly from the true mean of $p(x_0 \mid x_t)$ - meaning it becomes an inaccurate estimate. Since this mean is exactly the regression target provided to the neural network, an inaccurate target prevents the model from effectively learning the true mean of $p(x_0 \mid x_t)$, and consequently prevents it from accurately learning derived quantities such as the score and velocity.
>
> Why does the degenerate mean differ so drastically from the true $\mathbb{E}_{p(x_0 \mid x_t)}[x_0]$ in high-dimensional spaces? Our reasoning is as follows: high-dimensional data distributions are inherently complex(this is a common assumption in high-dimensional statistics), which results in a complex posterior $p(x_0 \mid x_t)$ . Approximating the mean using only a single sample is equivalent to performing Monte Carlo estimation with one sample. When $p(x_0 \mid x_t)$ has a complex structure, such a one-sample Monte Carlo approximation necessarily incurs large estimation errors.
>
>
> In Section 3.2, we analyzed two tasks and demonstrated that when $t < 600$,  the mean of $p(x_0 \mid x_t) $ tends to degenerate into a single sample. However, in practical applications, there is an additional factor that can lead to an even higher degree of degeneration. During training, although we use the empirical distribution as a proxy for the true data distribution, we do not compute $\mathbb{E}_{p(x_0|x_t)}[x_0]$ exactly. Instead, it is approximated using Monte Carlo methods, which inherently introduces a higher collapse ratio. The details are as follows:
>
> During training, we approximate the mean of $p(x_0|x_t)$ via Monte Carlo estimation, which fundamentally relies on sampling from $p(x_0|x_t)$. We implement this using Ancestral Sampling. Specifically, we first randomly select a sample $x_0$ from the dataset (equivalent to sampling from the empirical distribution $p_{\text{empirical}}(x) = \frac{1}{N-1} \sum_{i=0}^N \delta(x_0 - x_0^i)$), denoted as $x_0^\prime$, and then perform a noise addition process to obtain $x_t$ (equivalent to sampling from $p(x_t|x_0)$), denoted as $x_t^\prime$. This yields the paired sample $(x_0^\prime, x_t^\prime) \sim p(x_0, x_t)$, where $x_0^\prime \sim p(x_0|x_t = x_t^\prime)$. Repeating this procedure multiple times produces pairs $(x_0^\prime, x_t^\prime)^0, (x_0^\prime, x_t^\prime)^1, \dots, (x_0^\prime, x_t^\prime)^K$.
>
> While this approach allows us to obtain samples following $p(x_0|x_t = x_t^\prime)$, it cannot yield multiple samples for the specific $p(x_0|x_t = x_t^\prime)$. The reason is that, as the dimension increases, the space becomes vast, and it is highly unlikely that different pairs will produce similar or identical $x_t^\prime$. Consequently, the mean of each $p(x_0|x_t = x_t^\prime)$ can only be approximated using a single sample, which clearly introduces a significant estimation error.
>
> In general, as the dimension increases, data distributions become more complex. Combined with the above analysis, the degeneration phenomenon also becomes more severe. Consequently, the estimation error of the mean of $p(x_0|x_t)$ used during training grows larger. This strengthens our conclusion that, in high-dimensional settings, the model cannot effectively learn the true score or velocity (note that the dimension of modern generative tasks is rapidly increasing, such as in video generation).
>
> As for why the model can still ”work“ despite this issue—the reason lies in a different underlying mechanism, which is precisely what this paper aims to explain.

---

### Official Review · Reviewer_EE2E · 2025-11-01

**Soundness:** 1
**Presentation:** 1
**Contribution:** 1
**Rating:** 0
**Confidence:** 5

**Summary:**

This paper presents an analysis of diffusion models training objectives and its degradation to learning from a single sample in a low signa-to-noise regime. The paper also proposes a framework to unifies existing inference methods.

**Strengths:**

The paper proposed an interesting perspective to understand the training dynamics of diffusion models; in particular, how different frequences are picked up during different time and how the model's learning objective gradually shifts to focusing on a single sample as the signal to noise ratio increases. There are some empirical evidence in Table 1 that supports the authors' claim.

**Weaknesses:**

- The paper's language is very vague and the analysis not rigorous. A major claim in the paper is that diffusion models "cannot effectively learn the essential statistical quantities of the underlying data distribution, including the posterior, score, and velocity field", there are not rigorous mathematical arguments that support this claim. In fact prior work (e.g. [1]) have characterized the convergence properties of diffusion models. How would the proposed statement fit in this existing literatures on diffusion model theory?

- Many "supporting" arguments are not original and not surprising. For example, the "weighted sum degradation" behavior is expected: as the likelihood signal gets stronger, the posterior concentrates on the most consistent sample.

- The purpose of Section 4 on A UNIFIED INFERENCE FRAMEWORK -NATURAL INFERENCE is unclear. It appears more like an interpretation of existing inference methods. It is not clear what new  insights are drawn, or whether there are new inference methods that can be proposed and improve upon the existing ones.


References
[1] Chen, Sitan, et al. "Sampling is as easy as learning the score: theory for diffusion models with minimal data assumptions." arXiv preprint arXiv:2209.11215 (2022).

**Questions:**

See Weaknesses section.

---

> ### Author Response · Authors · 2025-11-15
> **First response to the Official Review**
>
> We sincerely thank the reviewer for the careful and thoughtful comments. Reviewing a theoretically oriented paper requires substantial time and effort, which we truly appreciate. Below, we provide point-by-point responses to the questions raised, and we hope that our discussion can help deepen the understanding of diffusion models.
>
> $&nbsp;$
>
> ## Responses to reviewer's question ##
> $&nbsp;$
>
> $\textbf{Question 1}$:
>
> >the paper's language is very vague and the analysis not rigorous. A major claim in the paper is that diffusion models "cannot effectively learn the essential statistical quantities of the underlying data distribution, including the posterior, score, and velocity field", there are not rigorous mathematical arguments that support this claim. In fact prior work (e.g. [1]) have characterized the convergence properties of diffusion models. How would the proposed statement fit in this existing literatures on diffusion model theory?
>
>
> $\textbf{Answer}$ :
> Here is how we derived this conclusion:
>
> First, in Section 2 -- Background of the paper, we show that the objective function of diffusion models essentially aims to fit the $\textbf{mean of the posterior distribution}$ $p(x_0 \mid x_t)$. In practical tasks, since the true data distribution $p(x_0)$ is unknown, the form of $p(x_0 \mid x_t)$ is also unknown. Without knowing the form of $p(x_0 \mid x_t)$, how can we provide the model with a target to fit (i.e., the mean of $p(x_0 \mid x_t)$ ) during training?
>
> We approximate it by using the $\textbf{empirical data distribution}$ instead of the true data distribution, which gives an approximate mean of $p(x_0 \mid x_t)$. In $\textbf{low-dimensional, densely sampled settings}$, the error between this approximate mean and the true mean is small, allowing the model to effectively learn the mean of $p(x_0 \mid x_t)$ corresponding to the true data distribution, and thus compute quantities like the Score or velocity. However, in $\textbf{high-dimensional, sparsely sampled settings}$, this approximate mean can have substantial errors. This implies that during training, we cannot provide the model with an accurate fitting target, and therefore, the model cannot reliably learn the true mean of $p(x_0 \mid x_t)$, resulting in inaccurate Score or velocity estimates.
>
> The convergence properties discussed in [1] rely on the assumption that the model can accurately estimate the Score of the true data distribution (as explicitly stated in the abstract). This very assumption is precisely what we question and challenge in this paper.
>
> $&nbsp;$
>
> $\textbf{Question 2}$:
>
> >The purpose of Section 4 on A UNIFIED INFERENCE FRAMEWORK -NATURAL INFERENCE is unclear. It appears more like an interpretation of existing inference methods. It is not clear what new insights are drawn, or whether there are new inference methods that can be proposed and improve upon the existing ones.
>
> $\textbf{Answer}$ :
> We know that existing inference methods all rely on a key assumption: that the model can accurately estimate the Score or Velocity of the true data distribution. However, as discussed in the previous response, due to data sparsity, high-dimensional diffusion models cannot reliably learn the Score or Velocity. This creates a fundamental tension, highlighting the need to reconsider the underlying mechanisms of current inference methods.
>
> Starting from this perspective, we conducted a detailed analysis of various inference methods and found that all of them can be expressed in a $\textbf{unified form}$, which we refer to as the Natural Inference framework. This framework has several elegant characteristics:
> - It involves multiple model prediction operations, each formulated as a prediction of $x_0$, which aligns with the degenerated form of the training objective.
> - Each intermediate $\hat{x}_t$ obtained during inference can be represented as a mixture of signal and noise, with mixing proportions consistent with the construction of $x_t$ during training.
> - It provides a simple and intuitive new perspective for understanding the inference process, without relying on any statistical assumptions. This new perspective will offers insights for designing new inference methods.
>
> Although we do not propose a new inference method in this paper, the framework provides a basis for potential optimization. Within this framework, there exists an infinite set of coefficient matrices that satisfy the requirements. Existing inference methods correspond to a particular choice (a specific coefficient matrix), which may not be optimal. Identifying the optimal configuration among this infinite set is an interesting direction for future research.
>
> $&nbsp;$
>
> In addition, in Appendix C.6, we provide a simple and intuitive example showing how a five-step Euler inference method can be expressed in the Natural Inference framework, which may help readers better understand the Natural Inference Framework.

---

> > ### Author Response · Authors · 2025-11-30
> > **Second response**
> >
> > $\textbf{Question 2}$:
> > > Many "supporting" arguments are not original and not surprising. For example, the "weighted sum degradation" behavior is expected: as the likelihood signal gets stronger, the posterior concentrates on the most consistent sample.
> >
> > $\textbf{Answer}$ :
> > When $t$ is very small (i.e., the added noise is very small), the mean of $p(x_0 \mid x_t)$ indeed degenerates to a single $x_0$. However, in high-dimensional space, there are two additional factors that can also cause the mean of $p(x_0 \mid x_t)$ to degenerate to a single $x_0$, even when $t$ is not very small: one is the sparsity of the training data, and the other is that during training, we do not compute the mean of $p(x_0 \mid x_t)$ exactly—instead, we approximate it using a Monte Carlo procedure (Ancestral Sampling). A detailed explanation can be found in our first response to Reviewer nswE.
> >
> > $&nbsp;$
> >
> > $\textbf{Question 3}$:
> > > The purpose of Section 4 on A UNIFIED INFERENCE FRAMEWORK -NATURAL INFERENCE is unclear. It appears more like an interpretation of existing inference methods. It is not clear what new insights are drawn, or whether there are new inference methods that can be proposed and improve upon the existing ones.
> >
> > $\textbf{Answer}$ :
> > The motivation for proposing Natural Inference comes from two considerations.
> > First, as pointed out in the previous response, in high-dimensional sparse spaces, the training objective has already degenerated into fitting $x_0$ from $x_t$. Therefore, based on the principle of matching training and inference, it is natural to ask whether existing inference methods can also be expressed in the same form.
> >
> > Second, as analyzed in our first response to Reviewer nswE, since the fitting target—the mean of $p(x_0 \mid x_t)$—has degenerated into a single $x_0$, this fitted target will exhibit a significant discrepancy from the true mean of $p(x_0 \mid x_t)$ corresponding to the real data distribution. Consequently, the neural network will not learn the true mean of $p(x_0 \mid x_t)$, nor will it learn the true score or velocity associated with the real data distribution.
> >
> > However, existing inference methods rely heavily on the assumption that the neural network can effectively learn the true score or velocity of the data distribution. This creates a clear contradiction. Hence, it becomes necessary to reinterpret the working mechanism of current inference methods from a different perspective, which constitutes the second motivation for proposing Natural Inference.

---

### Official Review · Reviewer_FcHU · 2025-11-03

**Soundness:** 2
**Presentation:** 2
**Contribution:** 1
**Rating:** 2
**Confidence:** 3

**Summary:**

This paper challenges the conventional probabilistic interpretation of diffusion models. The authors argue that, in high-dimensional and sparse data regimes, the objective of diffusion models “degrades” — instead of learning statistical quantities (posterior, score, velocity field), the model merely learns to predict the original clean sample $x_0$ from its noisy counterpart $x_t$.

The paper presents both analytical reasoning and empirical statistics to support this claim. Specifically:

* The authors analyze the posterior $p(x_0|x_t)$ and show that, in high-dimensional spaces, its mean (normally a weighted sum over all samples) collapses to a single dominant sample — the so-called **Weighted Sum Degradation** phenomenon.
* Empirical measurements on ImageNet-256 and ImageNet-512 show nearly 100% degradation for small $t$ ($t < 600$).
* Building on this, they propose a “Natural Inference Framework” that unifies multiple sampling algorithms (DDPM, DDIM, DPM-Solver, DEIS, etc.) as simple autoregressive signal combinations without invoking probabilistic concepts.

The authors claim this provides a new, intuitive, and non-statistical understanding of diffusion model training and inference.

**Strengths:**

* The new perspective is interesting

* The proposed “Natural Inference” formulation offers an intuitive, signal-processing-style view of sampling.

**Weaknesses:**

* The **main claim is overstated and unsupported**: posterior concentration does not invalidate the probabilistic formulation of diffusion models.
* The “Natural Inference Framework” is **only a re-expression** of known samplers in linear form, offering no new theory, algorithm, or empirical advantage.
* No experimental evidence demonstrates that this perspective improves understanding or performance.
* The argument conflates numerical concentration with conceptual failure of probabilistic mode

**Questions:**

* In Section 3.1, $p(x_0)$ is represented using an empirical distribution over training samples. This is not a common practice in diffusion literature. Could you clarify whether there are existing works that also adopt such an empirical formulation as the basis for theoretical derivation?

* While many deterministic formulations of diffusion models exist, they generally do **not** reject the probabilistic foundation. What is the motivation or added value of explicitly denying the probabilistic perspective?

* I agree that in modern machine learning there are cases where probabilistic reasoning can be dropped—for example, linear regression can be derived either from a Gaussian noise assumption or purely geometrically from minimizing MSE. However, methodological proposals should ideally be grounded in **simple and principled intuition**. The probabilistic formulation of diffusion models has such a foundation through *score matching*. Does your proposed “Natural Diffusion” framework have an equally simple and principled starting point?

---

> ### Author Response · Authors · 2025-11-15
> **First Respond to the Official Review**
>
> We sincerely thank the reviewer for the careful and thoughtful evaluation. Reviewing a theory-oriented paper requires substantial time and effort, and we truly appreciate your work. Below, we address each of your questions in turn. We hope that our responses can help deepen the understanding of diffusion models.
>
> $&nbsp;$
>
>
> ## Responses to reviewer's question ##
> $&nbsp;$
>
> $\textbf{Question 1}$:
>
> > In Section 3.1, is represented using an empirical distribution over training samples. This is not a common practice in diffusion literature. Could you clarify whether there are existing works that also adopt such an empirical formulation as the basis for theoretical derivation.
>
> $\textbf{Answer}$ :
> In practical applications, such as image generation, the true data distribution is unknown. All we have is a finite set of samples drawn from it—that is, the training data. Consequently, during training we always use the empirical distribution as a substitute for the true distribution.
>
> When the dimensionality is low and the data are dense, the empirical distribution can approximate the true distribution reasonably well. However, in high-dimensional and sparse settings, the empirical distribution no longer provides a good approximation. It is precisely this gap that motivates us to re-examine whether the model can genuinely learn the underlying true data distribution.
>
> $&nbsp;$
>
> $\textbf{Question  2,3}$:
> >While many deterministic formulations of diffusion models exist, they generally do not reject the probabilistic foundation. What is the motivation or added value of explicitly denying the probabilistic perspective?
> >I agree that in modern machine learning there are cases where probabilistic reasoning can be dropped—for example, linear regression can be derived either from a Gaussian noise assumption or purely geometrically from minimizing MSE. However, methodological proposals should ideally be grounded in simple and principled intuition. The probabilistic formulation of diffusion models has such a foundation through score matching. Does your proposed “Natural Diffusion” framework have an equally simple and principled starting point?
>
> $\textbf{Answer}$ :
> As explained in our previous response, in high-dimensional spaces the sparsity of data causes the empirical distribution to deviate significantly from the true data distribution. This discrepancy also manifests in the objective function of diffusion models. As described in Section~2 (Background), the essence of the diffusion objective is to fit the mean of $p(x_0 \mid x_t)$, which can be understood as a weighted sum over multiple samples of $x_0$. However, due to high-dimensional sparsity, this mean degenerates into a single $x_0$.
>
> It is generally believed that high-dimensional data distributions are highly complex, and therefore $p(x_0 \mid x_t)$ should also exhibit substantial complexity. Approximating its mean using only a single $x_0$ introduces a large error. This implies that during training, we are unable to provide an accurate fitting target for the model. Consequently, we argue that the model cannot reliably learn the true statistical properties of the data distribution, such as the score, the velocity.
>
> Additionally, there is an interesting coincidence. As discussed in Section~4 of the paper, many existing inference methods can also be uniformly expressed in the form of predicting $x_0$. This form closely corresponds to the degenerated training objective. For example, each $x_t$ in the inference process can also be expressed as a mixture of signal and noise, and the mixing ratio matches exactly the one used when constructing $x_t$ during training. Therefore, we believe that the inference process of diffusion models is operating under a different underlying mechanism.
>
> $&nbsp;$
>
> $\textbf{Comment 1}$:
> >The main claim is overstated and unsupported: posterior concentration does not invalidate the probabilistic formulation of diffusion models.
>
> $\textbf{Answer}$ : If necessary, we can phrase the main claim of the paper in a more moderate or cautious way.
>
> $&nbsp;$
>
> In addition, in Appendix C.6, we provide a simple and intuitive example showing how a five-step Euler inference method can be expressed in the Natural Inference framework, which may help readers better understand the Natural Inference Framework.

---

### Meta-Review · Area_Chair_TATL · 2025-12-24

**Summary:**

This is not a serious research contribution. Reviewers raised a number of objections which were not adequately addressed. None of the observations in this paper are new: the closed form for the empirical score function is well-known, and it does not contradict the fact that diffusion models can learn the distribution. The "unified framework" adds no new algorithmic innovation, nor are there any empirical results. The claims are strong yet indefensible. The writing is poor and does not adopt a proper scientific tone.

**Reviewer Concerns:**

All reviewer concerns are outstanding.

**Reviewer Scores:**

Reviewer scores would remain unchanged.

---

### Decision · Program_Chairs · 2026-01-26

Reject